# Miniature Inverted-Repeat Transposable Elements: Small DNA Transposons That Have Contributed to Plant *MICRORNA* Gene Evolution

**DOI:** 10.3390/plants12051101

**Published:** 2023-03-01

**Authors:** Joseph L. Pegler, Jackson M. J. Oultram, Christopher W. G. Mann, Bernard J. Carroll, Christopher P. L. Grof, Andrew L. Eamens

**Affiliations:** 1Centre for Plant Science, School of Environmental and Life Sciences, College of Engineering, Science and Environment, University of Newcastle, Callaghan, NSW 2308, Australia; 2School of Chemistry and Molecular Biosciences, The University of Queensland, St. Lucia, QLD 4072, Australia; 3School of Health, University of the Sunshine Coast, Maroochydore, QLD 4558, Australia

**Keywords:** angiosperms, transposable element (TE), miniature inverted-repeat transposable element (MITE), RNA-directed DNA methylation (RdDM), repeat-associated small-interfering RNA (rasiRNA), microRNA (miRNA), *MICRORNA* (*MIR*) gene evolution

## Abstract

Angiosperms form the largest phylum within the Plantae kingdom and show remarkable genetic variation due to the considerable difference in the nuclear genome size of each species. Transposable elements (TEs), mobile DNA sequences that can amplify and change their chromosome position, account for much of the difference in nuclear genome size between individual angiosperm species. Considering the dramatic consequences of TE movement, including the complete loss of gene function, it is unsurprising that the angiosperms have developed elegant molecular strategies to control TE amplification and movement. Specifically, the RNA-directed DNA methylation (RdDM) pathway, directed by the repeat-associated small-interfering RNA (rasiRNA) class of small regulatory RNA, forms the primary line of defense to control TE activity in the angiosperms. However, the miniature inverted-repeat transposable element (MITE) species of TE has at times avoided the repressive effects imposed by the rasiRNA-directed RdDM pathway. MITE proliferation in angiosperm nuclear genomes is due to their preference to transpose within gene-rich regions, a pattern of transposition that has enabled MITEs to gain further transcriptional activity. The sequence-based properties of a MITE results in the synthesis of a noncoding RNA (ncRNA), which, after transcription, folds to form a structure that closely resembles those of the precursor transcripts of the microRNA (miRNA) class of small regulatory RNA. This shared folding structure results in a MITE-derived miRNA being processed from the MITE-transcribed ncRNA, and post-maturation, the MITE-derived miRNA can be used by the core protein machinery of the miRNA pathway to regulate the expression of protein-coding genes that harbor homologous MITE insertions. Here, we outline the considerable contribution that the MITE species of TE have made to expanding the miRNA repertoire of the angiosperms.

## 1. Introduction

The angiosperms, plant species that produce flowers and seeds surrounded by a protective covering (i.e., a fruit), form the largest and most diverse phylum within the Plantae kingdom. The tremendous diversity of the angiosperms is the result of genetic variation, with nuclear genome size varying by more than 1000-fold across the approximate (~) 300,000 angiosperm species [1,2,3]. In the angiosperms, as with most other plant species and eukaryotes, variation in nuclear genome size correlates strongly with the degree of repetitive DNA content, content largely in the form of transposable elements (TEs): DNA sequences with the ability to amplify their sequences and to mobilize from their origin to a new chromosome position [1,4,5]. For example, TEs only comprise 3.0% of the ~80 megabase (Mb) nuclear genome of the simple aquatic bladderwort species *Utricularia gibba* (humped bladderwort), while in direct contrast, at least 90% of the ~17 gigabase (Gb) hexaploid genome of bread wheat (*Triticum aestivum*) is composed of TEs (Figure 1) [6,7]. Between these two extremes, further demonstration of the tight correlation between TE content and overall nuclear genome size across the angiosperms is provided by *Arabidopsis thaliana* (*Arabidopsis*: ~20% TEs in a ~135 Mb genome) [8]; *Brachypodium distachyon* (*Brachypodium*: ~30% TEs in a ~355 Mb genome) [9]; *Oryza sativa* (rice: ~40% TEs in a ~430 Mb genome) [10]; *Glycine max* (soybean: ~50% TEs in a ~ 1.1 Gb genome) [11]; *Gossypium hirsutum* (upland cotton: ~65% TEs in a ~2.3 Gb genome) [12]; *Zea mays* (maize: ~85% TEs in a ~2.4 Gb genome) [13]; and *Hordeum vulgare* (barley: >85% TEs in a ~5.3 Gb genome) [14] (Figure 1). However, even in angiosperm species with small genomes such as humped bladderwort, the high copy number of each TE distributed across their chromosomes strongly suggests that TEs have undergone repeated rounds of species-specific activation, proliferation, and movement during evolution [6,8,9,10].

Waves of TE expansion and contraction have resulted in dramatic differences in the overall architecture of the nuclear genomes of even closely related angiosperm species, with more recently developed comparative genomic approaches revealing the dramatic speed with which active TEs can drive such change. For example, due to TE activation and proliferation, the size of the genomes of upland cotton, maize, and the wild relative of cultivated rice, *Oryza australiensis*, have doubled in the last 5-million-year period [15,16,17]. An overall change to genome architecture resulting from TE activation and proliferation has also been reported within single angiosperm species, with the 22% difference in the estimated total size of the nuclear genomes of the modern inbred line of maize, B73, and the more ancient Mexican landrace, *Palomero toluqueño* (Palomero), shown to be primarily the result of TE activation and proliferation [13,18].

Further to forming a significant driver of the variation in total genome size across the angiosperms, TE amplification and/or movement can also result in much more subtle molecular effects. Many TE species show preferential insertion into regions of plant chromosomes that harbor a higher density of protein-coding genes, with the ‘open’ conformation of the chromatin surrounding such regions likely allowing TE access to chromosomal DNA upon its reinsertion [19,20]. TE insertion into the regulatory sequences of a protein-coding gene (Figure 2A) can alter the expression of the gene, with elevated, reduced, or complete loss of gene expression equally likely to result from TE movement (Figure 2B,D) [20,21]. Similarly, the insertion of a TE into the regulatory sequences of a protein-coding gene, or immediately adjacent to such regions of protein-coding loci, can produce an entirely new expression profile for these genes, as many plant TEs harbor *cis*-regulatory elements that direct their own transcriptional activity (Figure 2C) [22]. For example, a protein-coding gene that was not responsive to environmental stimuli may become responsive to an environmental stimulus post-proximal TE insertion due to the transcriptional regulatory elements housed by the TE [22]. Furthermore, the movement of members of some TE families has been shown to leave behind ‘genetic footprints’, with the added or removed nucleotides likely to cause a shift in the reading frame of a protein-coding gene, a shift that can result in the production of a dysfunctional or completely non-functional protein (Figure F–H) [23,24]. Therefore, host plants have developed elegant molecular mechanisms to mitigate the negative consequences of TE activity to ensure that those changes that remain post-TE movement are either neutral or only very mildly detrimental to the overall health of the host plant [25,26].

## 2. Angiosperm Transposable Elements: Type I and Type II Transposons

The TEs documented across the angiosperm species characterized to date can be divided into two broad groupings based on their mode of transposition [27,28]. Type I TEs, or retroelements, are RNA-based transposons that use a ‘copy-and-paste’ mechanism of transposition to produce a new copy of themselves that inserts at a different (usually proximal) chromosome position. Due to the copy number of the retroelement increasing with each round of transposition, Type I TEs comprise vast volumes of the nuclear genomes of many angiosperm species [29,30]. For example, the ~85% of the maize genome and the ~90% of the wheat genome that are composed of TEs are almost exclusively represented by the retroelement class of TE [29,30]. Type I TEs can be further divided into two main classes based on their DNA sequence characteristics and include (1) long terminal repeat (LTR) retroelements, which as their name suggests, are defined by long stretches of repetitive DNA sequence at their 5′ and 3′ termini, and (2) non-LTR retroelements, which are in turn further subdivided into long interspersed nuclear elements (LINEs) and short interspersed nuclear elements (SINEs) [31,32]. With longer lengths of 5000 to over 7000 base pairs (bp), LTR and LINE retroelements are predominantly autonomous elements, encoding all the protein machinery necessary to complete their transposition cycle. In brief, and using the LTR class as an example, a LTR retroelement harbors DNA sequences highly similar to those of the promoter regions of protein-coding genes of the host, which affords their recognition and use as a transcription template by the plant-encoded DNA-dependent RNA polymerase II (Pol II) [33,34]. The resulting intermediate transcript is then used as a template by the retroelement-encoded reverse transcriptase to generate a complementary strand of DNA, and subsequent to the action of other retroelement-encoded proteins, including an RNase H enzyme and an integrase enzyme, which mediate the integration of a new and now completely double-stranded DNA copy back into a new chromosomal location in the plant genome, the transposition cycle of the LTR retroelement is complete [35,36]. In contrast to LTRs and LINEs, SINE retroelements form the most abundant species of nonautonomous element in the plant nuclear genome [37,38]. The shorter length of SINEs of 100 to 700 bp does not allow for this subclass of Type I TE to encode for the protein machinery required for its own transposition, and therefore, SINEs rely on the protein machinery encoded by the host genome and other retroelements to mediate their movement via a copy-and-paste mechanism of transposition [37,38].

Type II TEs are classified as DNA-based transposons and move via a ‘cut-and-paste’ mechanism of transposition due to either a single- or double-stranded DNA version of the transposon being excised from the original chromosome position for reinsertion at a new location [39]. While ‘cut-and-paste’ implies a conservative mechanism of transposition, members of this TE class typically transpose from replicated DNA to actively replicating DNA to leave one copy at the original location plus an additional copy at the new location following DNA replication. As per Type I TEs, Type II TEs are either autonomous or nonautonomous elements depending on whether the DNA transposon encodes its own transposase or if it requires the transposase encoded by another Type II TE [36,38]. DNA transposons are flanked by inverted repeat sequences at their termini, and it is these sequences that are recognized by the transposase enzyme to cleave the transposon out of the chromosomal DNA prior to reinsertion of the excised sequence at a different position of the plant nuclear genome [36,38]. Across angiosperm genomes, the most common Type II TEs include members of the *Mutator-like element* (*MULE*), *hobo-Activator-Tam3* (*hAT*), and *CACTA* superfamilies [38,39]. The majority of Type II TEs detected in plant nuclear genomes are nonautonomous elements, which have only retained the minimal sequences required to continue to be recognized as TEs, as they are either truncated derivatives of once autonomous DNA transposons or they continue to harbor stretches of terminal repeat sequences that share enough similarity with the termini of autonomous elements to enable their recognition and transposition via the cut-and-paste mechanism of movement [35,36,37,38,39].

## 3. Miniature Inverted-Repeat Transposable Elements (MITEs): Short Nonautonomous Type II Transposons

Full-length Type II transposons are typically several thousand bp in length and contain a single, centrally positioned open reading frame (ORF) that encodes for the transposase enzyme required for the chromosomal movement of the TE and terminal inverted repeats (TIRs) at the 5′ and 3′ ends of the TE [35,36,37,38,39], sequences that define the DNA segment for mobilization. Members of the miniature inverted-repeat transposable element (MITE) subclass of DNA transposons are much shorter (~100–800 bp) nonautonomous deletion derivatives of once full-length DNA transposons that lack the centrally positioned ORF of full-length elements but have retained the inverted-repeats at their termini due to the essential requirement of the TIRs for recognition by a specific autonomous element generated transposase [1,40,41]. The characterization of a mutant allele of the maize *WAXY* (*Wx*) gene that encodes a starch granule-specific ADP glucose glucosyltransferase led to the identification of the first MITE: a short 128 bp sequence with 14 bp TIRs and 3 bp target site duplication (TSD) sequences at its 5′ and 3′ termini [42]. Sequence homology searches revealed that 11 other maize genes and a single barley locus also harbored sequences of highly similar length (~130 bp), matching structural features (TIRs and TSDs), and internal nucleotide composition (rich in A and T residues). However, the search failed to establish any significant degree of similarity between the identified MITE and any other maize TE that was known at the time [42]. Therefore, the researchers, Bureau and Wessler (1992), called the newly identified MITE family of TE the ‘*Tourist*’ family [42]. Two years after the discovery of the *Tourist* MITE family, the same researchers employed a computational approach to identify structurally similar short-length DNA transposons in grasses, which they termed the ‘*Stowaway*’ family of MITEs [43]. Following these initial discoveries, the MITE subclass of Type II TE has been identified and characterized in over 40 angiosperm species, including *Arabidopsis*, rice, barley, and wheat [44,45,46].

Considering that MITEs are internal sequence deletion derivatives of once full-length autonomous DNA transposons, the nucleotide composition of the remaining internal sequences and the similarity of the sequence of the TIR (length: ≥10 bp) and/or the TSD (length: 2–10 bp) at each end of a MITE have been used to group angiosperm-specific MITEs into families. To date, seven MITE superfamilies have been identified in the angiosperms by this approach and include the *PIF*/*Harbinger* (which includes *Tourist* MITEs), *Tc1*/*Mariner* (which includes *Stowaway* MITEs), *Mutator*, *Novosib*, *P element*, *PiggyBac*, and *hAT* superfamilies [22,40,47,48,49,50,51]. For example, a 3 bp TAA/TTA direct repeat forms the TSD of each *Tourist* MITE that belongs to the *PIF*/*Harbinger* superfamily, whereas each *Stowaway* MITE that belongs to the *Tc1*/*Mariner* superfamily has a TSD at each of its two ends that consists of a 2 bp TA/AT direct repeat [42,43]. Combined sequencing and bioinformatic approaches have now revealed that there are hundreds of MITE families present at high copy numbers in the nuclear genomes of numerous angiosperm species and, furthermore, that these elements are widely distributed throughout each genome. More specifically, the density of MITE insertions per chromosome has been shown to vary considerably; however, the overall pattern of MITE chromosome distribution does not appear to be random [22,52,53]. Surprisingly, despite the widespread prevalence of MITEs across the angiosperms, there does not appear to be a positive correlation between the total MITE copy number and the total number of TEs housed in the nuclear genome of a specific angiosperm species. For example, the 343,485 MITE-related sequences identified by Chen and colleagues (2013) were shown to represent less than 1.0% of the total combined genome size of the 19 different accessions of *Arabidopsis* assessed [54]. Similarly, MITEs were shown to only represent ~0.3% of the total TE fraction (~85%) of the maize genome [13], whereas in rice, the 179,415 MITEs, which were grouped into 339 families, were shown to account for over 5% of the total ~40% TE load of the rice nuclear genome [55]. Such a finding is consistent with the unique expansion or contraction of the total TE load that occurs during the evolution of each angiosperm species.

The widespread distribution of abundant MITE families across the chromosomes of characterized angiosperm species, and the high copy number of each identified MITE, suggest that this class of TE has potentially provided a potent resource for the molecular restructuring of the nuclear genomes of select angiosperm species as part of their continuing evolution [13,22,52,53,54,55]. Furthermore, and as outlined above, TE mobilization can result in altered gene expression or a change to the function of the protein product encoded by a gene [20,21,22,23,24]. Compared with other TE species, the activation and movement of the MITE class of TE potentially has a greater propensity to alter the expression profile and/or function of a gene due to the repeated demonstration that MITEs preferentially transpose within gene-rich regions of chromosomes [22,52,56,57,58]. Using the *Monkey King* species of *Tourist*-like MITEs as an example, positional analysis of *Monkey King* insertions on the chromosomes of *Brassica rapa* (field mustard) and *Arabidopsis* revealed that 74.4% of the 504 insertions detected in field mustard were within 3000 bp of a protein-coding gene, while 92.1% of the 38 *Monkey King* insertions detected in *Arabidopsis* mapped to within 1000 bp of a protein-coding gene [58].

Stemming from their preferential transposition within or adjacent to gene-rich regions of the genome, examples of the consequences of MITE movement in altering either standard plant development or the response of a plant to different forms of environmental stress have been detailed by numerous elegant studies across multiple angiosperm species, including *Pisum sativum* (pea) [59,60], *Arachis hypogaea* (peanut) [61,62], *Solanum tuberosum* (potato) [63], *Sorghum bicolor* (sorghum) [64,65], maize [66,67], wheat [68,69,70], and rice [71,72,73]. For example, one of the seven phenotypic traits used by Gregor Mendel to establish his fundamental rules of genetics, the wrinkled seedcoat phenotype displayed by pea plants, has been shown to be the result of MITE insertion into the promoter region of *STARCH-BRANCHING ENZYME1* (*SBE1*) [59,60]. More specifically, the insertion of a *hAT*-like MITE into the promoter region of the pea *SBE1* (*Psa-SBE1*) gene greatly reduced the total abundance of starch as well as the amylopectin to starch ratio in the early stages of embryo development, where in wild-type pea plants, which display smooth seedcoats, the *Psa-SBE1* gene is highly expressed [59,60]. As a result of defective starch production, the concentration of sucrose is much greater in early embryos of *sbe1* seeds, which leads to elevated water uptake by the mutant seeds during the initial stages of their development. However, the enhanced water uptake subsequently leads to much greater degrees of water loss as the seeds of *sbe1* mutant plants continue to mature, ultimately leading to the expression of a wrinkled seedcoat phenotype by mature *sbe1* seeds [59,60].

Importantly, MITE insertion has been reported to induce desirable phenotypic alterations in agronomically important angiosperm species. More specifically, together with FATTY ACID DESATURASE 2A (FAD2A), FAD2B is largely responsible for the conversion of oleic acid to linoleic acid in most vegetative tissues of the peanut plant [61], with vegetable oils high in oleic acid content over other polyunsaturated or saturated fatty acids being highly desired for use in either industrial applications or as an important supplement in the human diet [62]. The insertion of a MITE into the reading frame of the *Ahy-FAD2B* gene, which resulted in a frameshift mutation that led to the production of a truncated and non-functional *Ahy*-FAD2B protein, together with reduced *Ahy*-FAD2A protein abundance mediated by either natural (TE element insertion) or artificial means (chemically-induced mutation), was shown to be adequate to produce peanut lines that expressed high-oleate phenotypes [61]. This highly desired phenotypic trait led the authors to subsequently develop a DNA-based marker strategy to identify other peanut varieties that also contain MITE-directed alterations in either of the two peanut *FAD2* genes [61]. In contrast to the peanut work, where MITE insertion into a protein-coding gene was demonstrated to result in the expression of a desirable phenotype, the loss of a MITE insertion from the coding sequence of a potato gene has also been demonstrated to result in the expression of a desirable phenotype for the consumer market. An overall lower anthocyanin content in potato tubers with yellow-colored flesh and pale red to pink-colored skin has been shown to be the result of transposition of the *Stowaway* MITE, *dTstu1*, into the coding sequence of the anthocyanin biosynthesis pathway gene, *FLAVONOID 3′,5′-HYDROXYLASE* (*F3′5′H*) [63]. Potato tubers with dark red to purple colored flesh and skin, such as the tubers produced by the Jaga Kids Purple (JKP) variety (a potato variety popular with Japanese consumers), have been shown to express these darker color variations in the flesh and skin of tubers due to an elevated abundance of anthocyanin pigments stemming from increased *Stu-F3′5′H* enzyme activity post the transposition of the *dTstu1* MITE out of the first exon of the *Stu-F3′5′H* gene [63].

In addition to dicotyledonous (dicot) species, and potentially due to their typically larger sizes and DNA repeat-rich nuclear genomes, MITE-directed phenotypic alterations have also been reported in monocotyledonous (monocot) species, including several agriculturally important grass crop species. In sorghum, for example, polymorphism of the copy number of a *Tourist*-like MITE in the promoter region of a gene that encodes a putative multidrug and toxic compound extrusion (MATE) transporter as part of the *Alt_SB_* (*aluminum tolerance*, *Sorghum bicolor*) locus [64] has been linked to enhanced aluminum (Al) tolerance [65]. More specifically, screening a suite of sorghum lines revealed a strong positive correlation between increased *Tourist*-like MITE copy number in the promoter region of the putative *Sbi-MATE* gene, and enhanced tolerance to Al toxicity, with the authors going on to demonstrate that the sorghum lines that were tolerant to Al toxicity were able to excrete higher levels of citrate from their roots [65].

In maize and wheat, MITE movement has also been associated with either an accelerated or delayed onset of flowering [66,67,68,69]. In maize, MITE insertion upstream of the *Rap2.7* gene, a gene that encodes an APETELLA2 (AP2)-like transcription factor homolog, promotes the early transition from vegetative to reproductive development [66]. The *Zma-Rap2.7* gene belongs to the *Vegetative to generative transition 1* (*Vgt1*) locus, with Castelletti and colleagues (2014) revealing that MITE insertion into this locus leads to hypermethylation of the surrounding chromosomal landscape [66]. The enhanced degree of DNA methylation at the *Vgt1* locus upon MITE insertion leads to reduced *Zma-Rap2.7* gene expression, and in turn, the lower abundance of the *Zma*-Rap2.7 floral repressor protein drives the early transition of maize to flowering [66]. Delayed flowering has also been associated with MITE movement in maize [67], with the flowering time of male plants significantly delayed in lines that harbor a MITE insertion at the 3′ end of a gene encoding a CYTOCHROME P450 enzyme. The authors went on to identify small-interfering RNAs (siRNAs) with complementarity to the *CYTOCHROME P450* gene and to the inserted MITE from which the siRNAs were likely derived [67], a finding that suggests that siRNA-directed regulation of the expression of the *CYTOCHROME P450* gene, or of another gene with a considerable degree of sequence homology, is the likely cause of the observed delay to the onset of flowering in male maize plants. In wheat, the presence or absence of the MITE, *MITE_VRN*, in the promoter region of the *VERNALIZATION1* (*VRN1*) gene, which encodes an APETELLA1 (AP1) meristem identity-like protein, has been associated with differences in the vernalization response of winter and spring wheat varieties [68,69]. Interestingly, a more recent study [70] has further revealed that a microRNA (miRNA), *Tae*-miR1123, is also derived from *MITE_VRN*. The detection of a MITE-derived miRNA not only demonstrates that the DNA sequence of *MITE_VRN* is used as a template for transcription, but that post-transcription, the *MITE_VRN* transcript adopts a folding structure that forms a novel precursor transcript, which is processed to produce a novel member of the miRNA class of small regulatory RNA (sRNA).

A role for MITEs in hastening an evolutionary response to environmental pressures, has also been reported. In the rice cultivar Nipponbare and in other Japanese cultivars, the *mPing* MITE appears to be somewhat inactive, with only 50 copies detected; however, in the Chinese cultivar Ginbouzu, *mPing* has undergone considerable activation and proliferation, with over 1000 *mPing* insertions detected in the Ginbouzu nuclear genome [50,71]. Owing to its preferential insertion in gene-rich regions of rice chromosomes and with this species of MITE housing its own transcription regulating elements (i.e., *mPing* contains its own *cis*-regulatory elements), *mPing* movement and copy number expansion in Ginbouzu have been associated with conferring the observed tolerance of this cultivar to the environmental stresses of elevated salt, waterlogging, and low temperature [22]. The activation and movement of other MITE species in rice have also been associated with altering the phenotypic characteristics of leaf angle and seed size as well as modifying physiological properties such as the response of individual rice cultivars to plant hormones [72,73]. Moreover, as demonstrated in maize and wheat [68,71], the trait modifications reported to be the result of MITE activation and movement in rice appear to be under the regulation of sRNAs (miRNAs and siRNAs) derived from the MITEs themselves after their mobilization [72,73].

## 4. Repression of TE Activity by the RNA-Directed DNA Methylation (RdDM) Pathway, and TE Mechanisms to Avoid RdDM-Mediated Repression

Considering that only a small percentage of TE movement or proliferation events have been shown to result in a phenotypically advantageous outcome for the host that is positively selected for in subsequent generations [22,61,64,65], together with the demonstration that the vast majority of such events are at best selectively neutral, or at worst, lethal to the host [74,75], it is unsurprising that the angiosperms have developed complex molecular systems to render most of the TEs housed in their nuclear genomes in an inactive state [25,26]. RNA-directed DNA methylation (RdDM), directed by the 24 nucleotide (24-nt) repeat-associated siRNA (rasiRNA) class of sRNA, forms the primary molecular protection strategy employed by angiosperms to keep most of the TEs housed in their nuclear genomes in a repressed and inactive state [25,26,39,74]. The considerable volume of molecular resources that a plant devotes to the RdDM pathway to keep TEs in a repressed state is readily evidenced by sRNA sequencing datasets, which show that regardless of the cell or tissue type from which the global sRNA population was extracted, a significant proportion of the sRNAs that accumulate in plant cells belong to the 24-nt rasiRNA class of small regulatory RNA and possess complementarity to one or another of the major TE species housed by the host nuclear genome [76,77].

Largely depending on the sequence composition of the TE, together with its chromosomal and/or genomic position, either Pol II or the related and plant-specific DNA-dependent RNA polymerases, Pol IV and Pol V, are responsible for the transcription of a non-coding RNA (ncRNA) [78,79,80]. The Pol IV-generated ncRNA is recognized by the plant-encoded RNA-DEPENDENT RNA POLYMERASE2 (RDR2), which uses the ncRNA as a template for the synthesis of perfectly double-stranded RNA (dsRNA) (Figure 3A) [81,82]. The RDR2-generated dsRNA is then processed by the DICER-LIKE (DCL) endonuclease, DCL3, to produce 24-nt rasiRNA/rasiRNA* duplexes [81,82]. Next, each duplex is loaded into an ARGONAUTE (AGO) effector protein (primarily AGO4 homologs in most plant species), where the two strands unwind, leaving the mature and still AGO-loaded rasiRNA guide strand to form the catalytic core of the RNA-induced transcriptional silencing (RITS) complex (Figure 3A) [83]. The RITS complex uses the AGO-loaded rasiRNA as a sequence specificity guide to interact with other ncRNAs transcribed by the activity of Pol V, and after complex formation with a suite of histone modifying enzymes and DNA methyltransferases, such as DOMAINS REARRANGED METHYLASE2 (DRM2), the RITS complex directs the reinforcement of DNA methylation and other chromatin repressive modifications upon TE rich regions of angiosperm genomes to ensure that TEs remain in an inactive state (Figure 3A) [78,79,84].

More recently, it has been further shown that TEs that have managed to ‘escape’ gene silencing at the transcriptional level via the RdDM mechanism of RNA silencing or that have recently become activated to transpose to a new genomic position can have their ‘potential’ future activity inhibited at the posttranscriptional level. More specifically, RNAs transcribed from newly activated TEs can be recognized by RDR6 and used as a template for dsRNA synthesis [85,86]. The resulting dsRNA is processed into either 21- or 22-nt siRNA/siRNA* duplexes by DCL4 and DCL2, respectively. Post separation of the siRNA guide strands from their corresponding siRNA* passenger strands, the RNA-induced silencing complex (RISC) is guided by the AGO1-loaded 21- and 22-nt siRNAs to direct cleavage of any complementary RNA transcript [87,88]. Cleavage of any TE-derived transcripts by this alternate RDR6/DCL2/DCL4/AGO1-mediated pathway ensures that TE element activity is also repressed at the posttranscriptional level, in addition to TE activity being repressed at the transcriptional level via the canonical 24-nt rasiRNA-directed RdDM pathway [87,88]. Interestingly, in *Arabidopsis*, the initiation of the alternate RDR6-dependent RdDM pathway appears to rely on an initial miRNA-directed cleavage event of TE-derived transcripts for RDR6 recruitment to produce the dsRNA for the subsequent generation of 21- and 22-nt siRNAs by DCL4 and DCL2, respectively [87].

The importance of the rasiRNA-directed RdDM pathway to maintain TEs in an inactive state was elegantly demonstrated during the early molecular characterization of the pathway, which analyzed *Arabidopsis* mutant lines defective in the activities of core pieces of pathway protein machinery [89,90]. Specifically, in the *Arabidopsis* mutant lines *ddm1*, *met1*, and *rdr2*, defective in the activities of DECREASE IN DNA METHYLATION1 (DDM1), METHYLTRANSFERASE1 (MET1), and RDR2, respectively, reduced DNA methylation of TE sequences was associated with elevated transcriptional activity at TE loci and, for some specific species of TE, increased rates of transposition [89,90,91]. However, the degree of complexity and/or specificity of the RdDM pathway with respect to its ability to regulate TE movement has also been demonstrated, with certain species of TE having their level of activity reduced in the maize *rdr2* mutant [39,89] and not increased, as would be expected based on the *Arabidopsis* findings [89,90,91]. Other TEs, including certain members of the *Mutator* and *Suppressor-Mutator* TE families in maize, have developed other mechanisms to avoid having their transpositional activity suppressed by the rasiRNA-directed RdDM pathway [39]. More specifically, the transposase enzyme encoded by these *Mutator* and *Suppressor-Mutator* TEs can additionally function as a DNA demethylase enzyme (an enzyme that removes methyl groups from cytosine residues), further to the traditional and usually sole function of other transposases [39]. In addition, the capture of fragments of protein-coding genes appears to be a common occurrence for members of some TE superfamilies [55,92,93], a phenomenon that may have evolved to ‘confuse,’ and therefore evade the surveillance machinery proteins central to the mechanistic action of the RdDM pathway. Gene fragment capture by TEs, a process termed transduplication, has been well documented in rice and maize. In rice, for example, over 3000 *Pack-MULE* TEs have been shown to have captured and retained fragments from over 1000 protein-coding genes [74], and similarly in maize, an almost identical number of protein-coding gene fragments have been captured by numerous members of the *Helitron* TE family [94]. The capture of fragments of protein-coding genes appears to be an effective strategy adopted by TEs to evade the repressive effects of the rasiRNA-directed RdDM pathway, with a strong correlation revealed between the number of individual gene fragments harbored by a TE and the degree to which the TE is expressed [95].

Only a small percentage of the total number of MITEs identified to date across the angiosperms have been shown to form a source for rasiRNA production, which, after their functional maturation, directs an RdDM-mediated mechanism of chromatin silencing to repress the activity of the MITE itself. More specifically, 24-nt rasiRNAs are processed by DCL3 in an RDR2-independent manner from the relatively long, nearly perfectly self-complementary stem arms that form after the folding of the MITE-derived ncRNA [73,96]. The resulting MITE-derived rasiRNAs will then direct a canonical RdDM mechanism of RNA silencing to epigenetically modify the DNA landscape of target loci either in *cis* (i.e., the initially transcribed MITE) or in *trans* (i.e., highly homologous MITEs that belong to the same family as the initially transcribed MITE) [73,96]. In rice, for example, 24-nt MITE-derived rasiRNAs have been shown to differentially accumulate following the application of the abiotic stresses of cold, drought, and salt stress [97], and due to their preferential insertion in gene-rich regions of angiosperm chromosomes, 24-nt MITE-derived rasiRNAs have also been shown to add to the regulatory complexity of the expression of neighboring protein-coding genes [73]. Indeed, the strong preference of MITEs to move within gene-rich regions of angiosperm chromosomes has been proposed as a potential evasion strategy used by this class of TE to avoid repression by the rasiRNA-directed RdDM pathway. In *Brassica napus* L (rapeseed), for example, of the 677 *Monkey King* insertions revealed to be conserved across four assessed accessions, ~73% (*n =* 494) of insertions were positioned less than 5000 bp from a protein-coding gene, and of the 79 *Monkey King* insertions within gene bodies, only 19 (~24%) copies were positioned in exons, with the remaining 60 copies (~76%) inserted in the regulatory sequences (i.e., introns, or 5′ and 3′ untranslated regions (UTRs)) of the protein-coding genes [52]. Similarly, in *Phyllostachys edulis* (moso bamboo), the three most prevalent MITE families (including the *hAT*-like, *Mutator*-like, and *Stowaway*-like MITE families) all showed preferential insertion into gene-rich chromosome regions, and each member of these three MITE families was further shown to have a stronger preference for insertion into either the promoter region, 5′ UTR, 3′ UTR, or terminator region of a gene than into either the intron or exon of a gene [98]. The preferential insertion of MITEs into gene-rich regions of plant chromosomes may simply stem from the more ‘open’ or ‘relaxed’ conformation of the chromatin that surrounds such genomic regions, a conformational state that would more readily facilitate MITE access to chromosomal DNA upon reinsertion [19,20]. However, alternatively, the preferential positioning of MITE copies in such regions of the genome would ‘force’ the cell of the plant to decide whether the MITE sequence will be identified and used as a source of rasiRNA production, as the RdDM pathway directed by this class of sRNA could also potentially result in the epigenetic modification of the DNA and chromatin landscapes of neighboring protein-coding genes.

Another evasion strategy employed by MITEs to avoid the repressive effects imposed by the rasiRNA-directed RdDM mechanism of RNA silencing is the accumulation of nt alterations to the sequences internal to the highly conserved TIR and TSD sequences that define the 5′ and 3′ termini of each MITE. The continued accumulation of nt alterations to the internal sequence of a MITE would, over time, transform the dsRNA folding structure of a MITE-derived transcript from one of high sequence complementarity, which forms post-folding a perfect or near-perfect dsRNA molecule, to a stem-loop structured dsRNA with low degrees of complementarity between the two stem arms [99,100]. Such a conversion from a highly complementary dsRNA molecule to a stem-loop structured molecule with a low degree of dsRNA complementarity would ‘transition’ the MITE transcript from a substrate for rasiRNA production by DCL3 [82] to a molecule that very closely resembles the structure of a miRNA precursor transcript, which would instead be recognized and processed by DCL1, the plant DCL protein that is almost exclusively responsible for the production of the miRNA class of sRNA [101,102]. The preferential positioning of MITEs in gene-rich regions of the nuclear genomes of angiosperms [103,104] would potentially further prevent MITE loci from being recognized as a template for the synthesis of a ncRNA transcript for rasiRNA production. More specifically, chromosome regions rich in protein-coding gene sequences form the transcription template for messenger RNA (mRNA) synthesis by Pol II, whereas repeat-rich chromosome regions are more likely to form the transcription template for ncRNA synthesis by either Pol IV or Pol V as part of the rasiRNA-directed RdDM pathway [83]. As evidenced by the high copy numbers of most MITEs identified to date in angiosperms and the preferential insertion of MITE loci in gene-rich regions of chromosomes [103,104], together with the folding structure of MITE transcripts closely mimicking those formed by miRNA precursor transcripts [99,100], MITEs appear to have evolved effective evasion strategies to avoid the repressive effects of the rasiRNA-directed RdDM mechanism of RNA silencing.

## 5. The Plant microRNA Pathway

The miRNA class of sRNA was first identified in the dicot model species *Arabidopsis* in 2002 [105], with the authors of this pioneering study going on to demonstrate that many of the miRNAs identified in *Arabidopsis* have a high degree of conservation at the nt level to miRNAs present in the model monocot species rice [105]. This high level of miRNA sequence conservation has since been confirmed for other agriculturally important angiosperm species, including rapeseed [58], wheat [70,106,107], upland cotton [108], barley [109], maize [110], soybean [111], sorghum [112], and potato [113]. Largely stemming from technical advances in high-throughput sequencing technologies, many thousands of additional ‘species-specific’ miRNAs (also referred to as ‘young’ or ‘newly-evolved’ miRNAs) have also been identified in addition to the highly conserved miRNAs found across a wide range of unrelated plant species [58,108,109,110,111,112,113]. Both the conserved and newly-evolved miRNA subclasses have been shown to regulate the expression of genes essential for normal plant development [114,115,116], for a plant to respond to environmental stress [117,118,119], or to mount a defense response against viral, bacterial, fungal, and insect pathogens [120,121,122,123].

Based on the historical mechanistic findings made in the model dicot species *Arabidopsis*, the functional roles of the core pieces of protein machinery of the miRNA pathway are outlined as follows: the vast majority of plant miRNAs originate from *MICRORNA* (*MIR*) genes, with *MIR* genes sharing structural features with protein-coding genes, including a promoter and terminator sequence, which allows for the *MIR* gene to be used as a template by Pol II for the transcription of a ncRNA (i.e., the ncRNA does not harbor a translation start and stop codon) [124,125]. Each ncRNA transcribed from a *MIR* gene contains a region of partial self-complementarity, which directs the folding of the ncRNA into a stem-loop structured dsRNA termed the primary-miRNA (pri-miRNA) (Figure 3B) [126]. Post-folding, the pri-miRNA is recognized by the dsRNA binding protein, SERRATE (SE1), which binds the pri-miRNA for its transportation and presentation to DCL1 for the first round of precursor transcript processing [101,102,127,128]. A second dsRNA binding protein, dsRNA BINDING1 (DRB1), assists DCL1 at this processing step to ensure that DCL1 accurately converts the pri-miRNA transcript into a shorter length processing intermediate, termed the precursor miRNA (pre-miRNA) [129,130]. To ensure further processing accuracy, DCL1 together with DRB1 next converts the pre-miRNA, a short stem-loop structured molecule, into a miRNA/miRNA* duplex via cleavage of the stem arms and the loop region of the pre-miRNA (Figure 3B) [129,130,131]. DCL1 processing of pri-miRNAs and pre-miRNAs to generate miRNA/miRNA* duplexes leaves a characteristic cleavage signature of a 2-nt overhang at the 3′ terminus of both duplex strands [101,102]. Prior to separation of the miRNA guide strand from the miRNA* passenger strand, the sRNA-specific methyltransferase, HUA ENHANCER1 (HEN1), adds a methyl group to the 2′ OH group of the terminal 3′ nt of each duplex strand [132], a chemical modification that distinguishes all sRNA classes that accumulate in *Arabidopsis* cells from other RNA molecules of a similar size, such as RNA degradation products. The localization of AGO1 to both the nucleus and cytoplasm of *Arabidopsis* cells identifies AGO1 as the endonuclease likely responsible for separating the miRNA guide strand from the corresponding duplex strand, the miRNA* passenger strand [133,134]. Following its maturation, the miRNA guide strand is transported from the nucleus via either a HASTY (HST)-dependent or -independent exportation process (Figure 3B) [135]. In the cytoplasm of an *Arabidopsis* cell, the mature miRNA is loaded by AGO1 and used as a sequence specificity guide to direct repression of the expression of gene transcripts that harbor highly complementary target sites to the AGO1-loaded miRNA [133,134]. Due to the high degree of complementarity between each *Arabidopsis* miRNA and its targeted genes, in combination with the predominant positioning of target sites in the coding region of *Arabidopsis* target transcripts, plant miRNAs were originally assumed to exclusively regulate target gene expression via an AGO1-catalyzed transcript cleavage mode of RNA silencing; however, it is now widely accepted that translational repression also forms a mode of target gene expression repression directed by plant miRNAs (Figure 3B) [136,137].

In addition to its conserved miRNA population, a small cohort of newly-evolved miRNAs (miRNAs present in only a single or small number of closely-related plant species) also accumulate in *Arabidopsis* cells, including *Ath*-miR822, *Ath*-miR839, *Ath*-miR840, and *Ath*-miR869 [138]. In contrast to conserved miRNAs, which are processed from imperfectly complementary stem-loop-structured dsRNA precursors, newly-evolved *Arabidopsis* miRNAs are processed from precursors that adopt folding structures that more closely resemble inverted-repeats due to the high levels of complementarity between the 5′ and 3′ arms of the dsRNA [138]. Due to the unique dsRNA structure of the precursors of newly-evolved *Arabidopsis* miRNAs, DCL4 binds these precursors, and with the functional assistance of DRB4, it processes the precursor to generate the miRNA/miRNA* duplex [138,139,140]. In addition to DRB4, another DRB protein family member, DRB2, has also been shown to be required to produce this small cohort of *Arabidopsis* miRNAs [140]. It is worth mentioning here that DRB2 also appears to function together with DCL1, in place of DRB1, to produce other small subsets of conserved miRNAs in specific and developmentally important tissues [141,142]. However, besides the involvement of DCL4, DRB4, and DRB2 at this stage of the production pathway, it is highly likely that the functional roles played by other pieces of the core protein machinery of the conserved miRNA pathway, including the roles mediated by SE1, HEN1, HST, and AGO1, remain unchanged for the production of the DRB4- and DRB2-dependent subsets of *Arabidopsis* miRNAs.

## 6. Plant *MIR* Gene Evolution Models

Many miRNAs, especially conserved miRNAs, belong to multimember gene families, from which a nearly identical mature miRNA sequence is processed from structurally distinct precursors after the folding of the initially transcribed ncRNAs, which were expressed in a spatiotemporal pattern distinct from the expression domain of each of the other family members. Further, each individual locus of a multimember *MIR* gene family can either be evenly dispersed across the different chromosomes of a species, as reported for the twelve members of the rice *MIR156* gene family, or clustered together on select chromosomes, as has been demonstrated for the six and twenty members that comprise the *MIR395* gene family in *Arabidopsis* and rice, respectively [143,144]. Taken together, (1) high degrees of mature miRNA sRNA nt sequence conservation across different angiosperm species, (2) gene families composed of high member numbers, (3) the spatiotemporal expression pattern displayed by individual gene family members, and (4) the unique chromosome positioning of each member of a gene family, show that angiosperm *MIR* gene families have undergone considerable evolutionary diversification since the divergence of monocots and dicots ~200 million years ago [145]. Assessment of the member number of each *MIR* gene family detected across 10 species belonging to the *Oryza* genus has provided further evidence of the considerable evolutionary diversification of this specific gene class, with this analysis suggesting that the average number of members per *MIR* gene family is considerably higher in plants (*n =* ~2.5 members per family) than it is in animals (*n =* ~1.7 members per family) [146], a finding which suggests that different levels of gene duplication events have made a significant contribution to the genetic diversity of the miRNA repertoire of plants. The *de novo* origination of a plant *MIR* gene requires two fundamental steps, those being: (1) the generation of a DNA sequence that, when transcribed, produces a ncRNA that forms a folding structure that is recognized by the protein machinery of the miRNA pathway, and (2) the acquisition of a promoter region to enable the active transcription of the miRNA precursor transcript encoding sequence [147,148]. To date, the *de novo* formation of *MIR* gene promoters has not been well studied in plants; however, the *de novo* origination of pri-miRNA and pre-miRNA-forming sequences has been more extensively investigated across a range of angiosperm species by numerous research groups. Based on the findings reported by these studies, three angiosperm *MIR* gene evolution models have been proposed and include: (1) target gene duplication (Model 1), (2) random hairpin RNA forming structures (Model 2), and (3) transposed TE-derived (Model 3) models.

In *Arabidopsis*, early sequence similarity searches revealed that the transcript sequences for several pri-miRNA and/or pre-miRNAs, which flank either the miRNA or miRNA* sequence, share similarity with regions outside of the miRNA binding sites of their corresponding target gene transcripts [138,149,150,151]. For example, the precursor transcripts of the *Arabidopsis* miRNAs, *Ath*-miR161, *Ath*-miR163, *Ath*-miR822, *Ath*-miR839, and *Ath*-miR869, share similarity to the nt composition of their respective *PENTATRI-COPEPTIDE REPEAT* (*PPR*), *S-ADENOSYLMETHEONINE-DEPENDENT METHYLTRANSFERASE* (*SAMT*), *DIVERGENT C1* (*DC1*) domain, *P-GLYCOPROTEIN* (*P-GP*), and *SU*(*VAR*)*3-9*-like target genes [149,150]. These findings led to the proposal of *MIR* gene evolution Model 1: inverted duplication of a founder gene (which subsequently becomes the miRNA target gene) with or without the promoter of the duplicated founder gene [152]. Post such a duplication event, and if the newly duplicated sequence is transcribed, the folding of the initially formed transcript would generate a highly complementary dsRNA template for the production of many sRNAs (i.e., siRNAs) via the endonuclease activity of either DCL2, DCL3, or DCL4 (Figure 4A) [138,149,153]. Over time, the continued accumulation of nt alterations to key regions of the stem-loop, including the two stem arms and the loop region, would transition the dsRNA to a processing template for the production of a reduced number of miRNA-like siRNAs by DCL4 (such as the DCL4 processing patterns documented for the newly-evolved *Arabidopsis* miRNAs *Ath*-miR822, *Ath*-miR839, *Ath*-miR840, and *Ath*-miR869) and further modification of the precursor transcript encoding sequence would eventually convert the dsRNA to a bona fide stem-loop structured miRNA precursor transcript solely recognized by DCL1 for the production of a single, predominant sRNA species, the miRNA (Figure 4A) [138,149,150,154]. The further acquisition of transcription-regulating sequences by the newly formed *MIR* gene, together with the developmental importance of the target gene(s) of the now DCL1-dependent miRNA, would place selective pressure on the newly formed *MIR* gene locus to retain its genetic features at the chromosome level in subsequent generations. For example, an elegant 2019 study by Shanfa Lu on ten species of the *Vitis* genus, including *Vitis vinifera* (common grapevine), showed that all identified *Vvi-MIR1444* and *Vvi-MIR12112* loci originate from a common ancestral *POLYPHENOL OXIDASE* (*PPO*) gene targeted for posttranscriptional regulation by *Vvi*-miR1444 and *Vvi*-miR12112 [147]. The author went on to further demonstrate that the promoter sequences that drive the transcriptional activity of each of the multiple members of the *MIR1444* and *MIR12112* gene families are also derived from a region towards the 3′ end of the same ancestral *VviPPO* gene [147].

*MIR* gene evolution Model 2 (Figure 4B) proposes that random hairpin RNA (hpRNA) forming structures from (1) protein-coding gene introns, (2) other intergenic regions of angiosperm chromosomes that have acquired promoter activity for their transcription, or (3) DNA repeat rich regions of plant chromosomes, such as where multiple copies of the same autonomous LTR retroelement lie immediately adjacent to one another in opposite orientations, have given rise to the generation of new *MIR* genes [155,156]. Model 2 largely stems from studies conducted in *Arabidopsis*, namely, the comparison of orthologous regions of *A. thaliana* and *A. lyrata* chromosomes revealed limited sequence conservation of *A. thaliana MIR* genes in *A. lyrata* [157]. In addition, the encoding loci of a number of newly-evolved *Arabidopsis* miRNAs, including *Ath*-miR774, *Ath*-miR775, *Ath*-miR776, *Ath*-miR779, *Ath*-miR823, *Ath*-miR830, *Ath*-miR858, *Ath*-miR864, *Ath*-miR865, and *Ath*-miR870, fail to display any degree of similarity to other *Arabidopsis* genomic sequences [155]. Therefore, *MIR* gene evolution Model 1 cannot adequately account for the origin of such *MIR* genes. However, as stated for *MIR* gene evolution Model 1 (Figure 4A), Model 2 similarly proposes that post initial hpRNA formation, nt alterations subsequently accumulate in the hairpin encoding sequence to transition the folded dsRNA structure away from being used as a template for siRNA production by DCL2, DCL3, or DCL4 [138,149,153], to a stem-loop structured dsRNA molecule that is solely recognized by DCL1 as a processing substrate for miRNA production (Figure 4B) [138,149,150,154].

Taking aspects of both *MIR* gene evolution Model 1 and 2, Model 3 proposes that any region of a plant chromosome that has undergone inverted duplication can potentially form the starting point for the generation of a new *MIR* gene, with an early bioinformatic study identifying ~134,000 and ~410,000 imperfect inverted-repeats across the chromosomes of *Arabidopsis* and rice, respectively [158]. This study further identified the MITE class of TE to be a rich source of imperfect inverted-repeats in both the *Arabidopsis* and rice genomes due to the (1) small size of each species of MITE (generally ~100 to 800 nt in length), (2) the high copy number of each MITE species per genome, and (3) the presence of short stretches of highly conserved inverted repeat and target site duplication sequences at the 5′ and 3′ termini of a MITE [158]. Taken together, these genetic features have a high potential to form the raw material required for hpRNA and/or stem-loop formation post-MITE co-transcription, with either a proximally positioned protein-coding gene or an autonomous TE (Figure 4C) [99,100,158]. As proposed for *MIR* gene evolution Models 1 and 2 (Figure 4A,B), Model 3 also posits that the first dsRNA structure to form would initially be recognized as a processing substrate for 21-, 22-, and 24-nt siRNA production by DCL4, DCL2, and DCL3, respectively. Next, the ongoing modification of the initial dsRNA by a further nt alteration will subsequently direct precursor processing by a DCL4 for the production of a reduced number of miRNA-like siRNAs, and subsequently, via the introduction of even further nt alterations to the internal sequences of the molecule, the final transition of the folded transcript into a stem-loop structured RNA, which displays all of the typical features of a plant pri-miRNA and/or pre-miRNA for exclusive processing by DCL1 for the production of a newly-evolved miRNA (Figure 4C) [138,149,150,154].

Alignment of miRNA sRNAs to all classes of TE harbored by the rice nuclear genome revealed that ~80% of TE-mapped miRNAs aligned to the MITE class of TE, while only 10% and 9% of TE-derived miRNAs mapped to retroelements or another class of DNA TE, respectively [100]. This bioinformatic assessment readily demonstrated that the MITE class of TE has formed a rich source of newly-evolved *MIR* gene loci in this model monocot species [100]. Moreover, in rice, bioinformatic analyses have further revealed that the rice miRNAs *Osa*-miR437, *Osa*-miR443, *Osa*-miR812, *Osa*-miR814, *Osa*-miR816, *Osa*-miR818, *Osa*-miR1862, and *Osa*-miR5788 are all derived from *Stowaway*-like MITEs (members of the *Tc1*/*Mariner* MITE superfamily) [55,56], while *Osa*-miR439, *Osa*-miR2122, *Osa*-miR2877, *Osa*-miR5149, *Osa*-miR5153, and *Osa*-miR5827 were all mapped to members of the *Mutator* MITE superfamily [146,159]. Mapping of TE-derived miRNAs in other monocots and dicots has further identified the specific MITE superfamilies that have formed a rich source of *MIR* gene loci in plants. For example, in the monocots rice, *Brachypodium,* and *Setaria italica* (foxtail millet), ~40% of MITE-derived miRNAs mapped to the *Tc1*/*Mariner* superfamily, whereas ~50% of MITE-derived miRNAs were revealed to have originated from members of the *PIF*/*Harbinger* superfamily (which includes the *Tourist* MITE family) in two other monocots, specifically maize and sorghum [148]. In comparison, the *Mutator* MITE superfamily was revealed to be the source of ~60% of MITE-derived miRNAs in nine dicot species (including *Arabidopsis*, *A. lyrata*, *Citrus sinensis* (sweet orange), *Fragaria vesca* (wild strawberry), potato, *Prunus persica* (peach), *Solanum lycopersicum* (tomato), soybean, and *Medicago truncatula* (barrelclover)) [148]; furthermore, in the 15 angiosperm species assessed in the same study, ~71% of MITE-derived miRNAs were determined to have originated from members of only three MITE superfamilies, including the *PIF*/*Harbinger*, *Mutator,* and *Tc1*/*Mariner* superfamilies [148].

## 7. MITEs as a Source of *MIR* Genes in Angiosperms

Early bioinformatic studies attempting to identify and/or annotate either actively evolving or newly-evolved plant miRNAs with sequence similarities to TEs could have entirely overlooked such miRNAs or incorrectly classed TE-derived miRNAs as a species of DNA repeat-derived siRNA (i.e., incorrect classification of a bona fide miRNA as a rasiRNA due to its size and/or genome origin). However, with the now widespread acceptance of *MIR* gene evolution Model 3 (Figure 4C), TE-derived miRNAs have been reported for many angiosperm species and characterized in *Arabidopsis*, upland cotton, rapeseed, grapevine, wheat, and rice. Furthermore, the Plant Transposable Element-related microRNA Database (PlanTE-MIR DB; http://bioinfo-tool.cp.utfpr.edu.br/plantemirdb/ accessed on 1 February 2023) was originally developed for public use in 2016 and reported on the identification of 152 TE-derived miRNAs in nine angiosperm species, including *Arabidopsis*, barrelclover, *Brachypodium*, common grapevine, *Populus trichocarpa* (black cottonwood), potato, soybean, sorghum, rice, and the bryophyte *Physcomitrella patens* (spreading earthmoss) [160]. In 2018, the PlanTE-MIR DB dataset was updated as part of the release of the more expansive Plant Non-Coding RNAs related to TEs (PlaNC-TE: http://planc-te.cp.utfpr.edu.br/ accessed on 1 February 2023) database that has increased the number of identified TE-derived miRNAs in the same ten assessed plant species by almost 2-fold (*n =* 271) [161]. These online resources [160,161], together with other early bioinformatic analyses [99,100], have shown that the contribution of TE-derived miRNAs to the complete miRNA profile of each plant species differs widely and that the reported difference may be more likely due to the number of MITEs housed by the nuclear genome of an individual species than the total TE load of a species. For example, two early studies of MITE-derived miRNAs in *Arabidopsis* and rice suggested that only a handful (*n =* 5) of *Arabidopsis* miRNAs were derived from MITEs, whereas in rice, almost 75 *MIR* genes were shown to overlap with MITEs [99,100]. For context, ~20% of the *Arabidopsis* nuclear genome is composed of TEs versus the ~40% TE load of the rice nuclear genome [8,10], however, the rice nuclear genome harbors many more MITE families, with each family composed of many more copies than what has been reported for the MITE landscape of *Arabidopsis* [54,55]. This comparison again emphasizes the species-specific nature of the expansion and contraction of each class of TE over evolutionary time.

Alignment of *Arabidopsis* miRNA precursor transcripts and MITE genomic sequences has identified *Ath*-miR401, *Ath*-miR405a, *Ath*-miR405b, *Ath*-miR405c, *Ath*-miR405d, *Ath*-miR407, *Ath*-miR416, *Ath*-miR854a, *Ath*-miR854b, *Ath*-miR854c, *Ath*-miR854d, *Ath*-miR855 and *Ath*-miR1888 as the 13 *Arabidopsis* miRNAs where a MITE has potentially played a role in the formation of each corresponding *MIR* gene locus [99,150,160,161]. Furthermore, these analyses have identified members of the *hAT* and *Mutator* MITE superfamilies as the main source of MITE-derived miRNAs in this model dicot [99,150,160,161]. Of the 13 MITE-derived miRNAs putatively identified in *Arabidopsis* to date, the *Ath*-miR854 and *Ath*-miR855 sRNAs are the most thoroughly characterized, with both miRNAs shown by northern blot analysis to fail to accumulate to detectable levels in the *dcl1*, *drb1,* and *hen1* mutants, *Arabidopsis* plant lines defective in the activity of the core miRNA pathway machinery proteins DCL1, DRB1, and HEN1, respectively [161]. Further support that the *Ath*-miR854 and *Ath*-miR855 sRNAs function as bona fide miRNAs and not as repeat-derived rasiRNAs was provided by the same authors via their demonstration that (1) both sRNAs accumulated to wild-type equivalent levels in the tissues of the *Arabidopsis rdr2* mutant, a plant line defective for RDR2 activity, a core piece of RdDM pathway protein machinery, and (2) *Ath*-miR854 and *Ath*-miR855 regulate the expression of their shared target gene, *OLIGOURIDYLATE BINDING PROTEIN1b* (*Ath-UBP1b*), at the post-transcriptional level via directing a translational repression mode of RNA silencing (Figure 5) [162].

The *MIR482*/*MIR2118* (*MIR482*/*2118*) locus forms a unique plant *MIR* gene as it encodes a precursor from which two functional miRNAs, miR482 and miR2118, are processed [163,164]. *MIR482*/*2118* forms an ancient, highly conserved *MIR* gene family, with both miR482 and miR2118 sRNAs shown to regulate their numerous *NUCLEOTIDE-BINDING SITE LEUCINE-RICH REPEAT* (*NBS-LRR*) disease resistance target genes at the posttranscriptional level via a mRNA cleavage mode of RNA silencing (Figure 5) to maintain disease resistance balance in both gymnosperm and angiosperm species [163,164,165]. Via detailed molecular assessment of four species of the *Gossypium* genus, including the two tetraploid species, upland cotton and *G. barbadense* (Gallini cotton), and their diploid progenitors, *G. arboretum* (tree cotton) and *G. raimondii* (Peruvian cotton), Shen and colleagues (2020) revealed considerable expansion of the *MIR482*/*2118D* locus in upland cotton, Gallini cotton, and Peruvian cotton, but not in tree cotton [108]. In this elegant study, the authors went on to show that expansion of the *MIR482*/*2118D* locus in these three *Gossypium* species was the result of considerable transpositional activity of the *PIF*/*Harbinger* MITE, *GosTE* [108]. More specifically, 10, 18, and 22 additional copies of the *MIR482*/*2118D* locus were attributed to repeated rounds of transposition of the *GosTE* MITE in upland, Gallini, and Peruvian cotton, respectively [108]. Of further interest, the detailed bioinformatic analyses presented by Shen et al. (2020) also revealed nt alterations to the miR482d and/or miR2118d sRNAs processed from the precursors encoded by some of the newly-evolved *MIR482*/*2118D* loci, a miRNA sequence modification proposed to have occurred to expand the *NBS-LRR* target gene repertoire of these miRNAs as well as to demonstrate dynamic co-evolution of miR482/2118d and its target genes [108].

In the Brassicaceae family, *BraSto*, a *Stowaway*-like MITE, was the first MITE identified in *Brassica* species and has been demonstrated to continue to remain active in the gene-rich regions of both diploid and allotetraploid (hybrid species with four times the chromosome copy number of haploid species) family members [166]. In addition to *BraSto*, the *Tourist*-like MITE, *Monkey King*, has also been demonstrated to remain active across *Brassica* species, including field mustard and rapeseed [52,58]. Dai et al. (2015) used the *Monkey King* sequence to interrogate miRBase, the microRNA database (https://www.mirbase.org/ accessed on 1 February 2023), an approach that revealed perfect homology to the rapeseed miRNA, *Bna*-miR6031 [58]. Further computational analysis by Dai and colleagues (2015) showed that the *Monkey King* sequences that flank *Bna*-miR6031 and its corresponding *Bna*-miR6031* sequence were able to fold into a structure that perfectly mimicked the folding structures formed by the precursor transcripts of numerous other rapeseed miRNAs [58]. The authors also showed that *Bna*-miR6031 is a 24-nt miRNA and not the canonical length of 21-nt for most plant miRNAs, and furthermore, that localized DNA methylation changes occurred at gene sequences that harbor *Monkey King* insertions (Figure 5), thus forming potential targets of *Bna*-miR6031-directed expression regulation via a transcriptional-level mode of RNA silencing [58]. One such rapeseed gene that has been shown to harbor a *Monkey King* insertion in its gene body is *Bna-FLC.A10*, the rapeseed homolog of the *Arabidopsis* floral repressor protein, FLOWERING LOCUS C (*Ath*-FLC). Interestingly, in a subsequent study [52], *Bna*-miR6031 was shown to be differentially expressed (1) in different rapeseed tissues (including young leaves, old leaves, and roots) and (2) following the exposure of rapeseed plants to a range of environmental stresses (including cold, drought, heat, and waterlogging stress). Taken together, these findings suggest that at different stages of development or after the exposure of rapeseed plants to different environmental stresses, the 24-nt MITE-derived *Bna*-miR6031 sRNA potentially alters the expression of protein-coding genes that harbor *Monkey King* insertions in their gene bodies via directing a RdDM-based mode of transcriptional gene silencing (Figure 5) [52,58].

As outlined in the *MIR* gene evolution Model 1 section above (Figure 4A), in common grapevine and in nine other species that also belong to the *Vitis* genus, via extensive bioinformatic assessment, members of the *MIR1444* and *MIR12112* gene families have been shown to have evolved from the inverted duplication of an ancestral *PPO* gene, with multiple *PPO* genes now forming target genes for miR1444- and miR12112-directed expression regulation in these *Vitis* species [148]. In common grapevine, a third miRNA, termed *Vvi*-miR058, which has not been entered into the miRBase database to date [167], has also been demonstrated to have evolved from the same ancestral *PPO* gene as have *Vvi-MIR1444* and *Vvi-MIR12112* family members, and to regulate the expression of specific *Vvi-PPO* genes at the posttranscriptional level via a mRNA cleavage-based mode of RNA silencing [147]. Although a MITE was not demonstrated to have been involved in the evolution of the multiple members of the *MIR058*, *MIR1444,* and *MIR12112* gene families in *Vitis* species, transposition of a MITE is likely to have played a role in the evolution of the promoter region that now drives the transcriptional activity of each member of these three *MIR* gene families. More specifically, the common grapevine *MIR058*, *MIR1444,* and *MIR12112* genes have all been shown to be copper (Cu)-responsive genes due to the presence of multiple copies of Cu-Responsive Elements (CuREs: short DNA sequence motifs to which transcription factors bind to modulate gene expression in response to Cu levels) housed in the promoter region of each Cu-responsive *MIR* and protein-coding gene [147]. The extensive bioinformatic analyses reported in [147] go on to reveal that these *cis*-regulatory elements were likely ‘captured’ by a MITE following its transposition in to and out of the region that surrounds the stop codon of the coding sequence at the 3′ end of the same ancestral *PPO* gene from which the *MIR058*, *MIR1444,* and *MIR12112* genes in *Vitis* species are likely to have also evolved, findings that form another elegant demonstration of the dynamic co-evolution of miRNAs and their target genes in plants.

As outlined above, the presence or absence of the *Mutator*-like element, *MITE_VRN*, in the *Tae-VRN1* promoter region has been associated with the difference in the vernalization response displayed by some winter and spring varieties of wheat [68,69]. Use of the *MITE_VRN* sequence to query all wheat miRNAs stored in the miRBase database revealed a high degree of homology between a 23-nt section of the *MITE_VRN* sequence and the 23-nt wheat miRNA, *Tae*-miR1123 [70]. Yu and colleagues (2014) went on to confirm that *MITE_VRN* forms a novel source for miR1123 production in wheat via the demonstration that (1) the *MITE_VRN* transcript folds to form a stem-loop structure highly similar to the folding structures adopted by the precursor transcripts of other newly-evolved miRNAs, and via the use of northern blot hybridization analysis (2) both *MITE_VRN* and *Tae*-miR1123 were shown to differentially accumulate in different wheat varieties, at different stages of development of each assessed variety, and under differing growth regimes (including low temperature and short days) [70]. However, a positive relationship between *Tae-VRN1* gene expression and *Tae*-miR1123 abundance was revealed in the same study, with levels of both ncRNAs increasing with plant age and decreasing under low temperature and short days [70]. This finding strongly infers that the *MITE_VRN* insertion in the *Tae-VRN1* gene promoter region does not form a direct target for *Tae*-miR1123 expression regulation in wheat, and moreover, that *Tae*-miR1123 is more likely to simply represent a by-product of *Tae-VRN1* gene expression. In addition to *MITE_VRN* and *Tae*-miR1123, *Stowaway*-like MITEs of the *Tc1*/*Mariner* superfamily have also been shown to have played a role in directing the expansion of the *Tae-MIR1127*, *Tae-MIR1137,* and *Tae-MIR1436* gene families in wheat [106]. An additional bioinformatic analysis by Poretti et al. (2020) revealed that, like many other newly-evolved *MIR* genes, additional *MIR1127*, *MIR1137,* and *MIR1436* gene family members could only be identified in other species closely related to wheat, which also belong to the *Triticeae* tribe. A more recent study in wheat mapped 38 of the 270 (~14%) putative miRNAs identified in this species to a MITE origin, and furthermore, 28 of the 38 (~75%) MITE-derived miRNAs were 21-nt in length and not 24-nt, the predominant length of MITE-derived miRNAs in rice [107]. Interestingly, this finding suggests that wheat and rice may have differing DCL requirements for MITE-derived miRNA production, specifically: DCL1 is the primary DCL enzyme required for MITE-derived miRNA production in wheat, whereas in rice, DCL3 would occupy this functional role. The authors, Crescente et al. (2022), went on to perform positional analysis for the stem-loop encoding sequences of the 38 MITE-derived miRNAs, a mapping exercise that revealed that 17 (~45%) were located in transcriptionally active sites such as the promoter regions and introns of protein-coding genes, while the stem-loop encoding sequences of the remaining 21 (~55%) MITE-derived miRNAs were mapped to intergenic regions of gene-rich regions of wheat chromosomes [108]: MITE positioning that would potentially allow for the encoding sequences of all 38 identified MITE-derived miRNAs to be used as a template for transcription.

The rich contribution of MITEs to the global miRNA population of rice was identified by an early study attempting to uncover the contribution of all TE species to the total miRNA landscapes of *Arabidopsis* and rice [99,100]. More specifically, the findings of these studies suggested that only five miRNAs potentially originated from a MITE in *Arabidopsis*, whereas in rice, 75 miRNAs were mapped to a MITE origin [99,100]. These findings showed that although the rice nuclear genome is only slightly over three times the size of the *Arabidopsis* nuclear genome [8,10], the MITE species of Type II TE have made a 15-fold greater contribution to *MIR* gene evolution in rice than the contribution made by MITEs to the total miRNA population of *Arabidopsis* [99,100]. As stated above, rice MITE-derived miRNAs are frequently 24-nt in length and not the canonical size of 21-nt for conserved rice miRNAs. It is important to note here however, that an equivalent abundance of 21- and 24-nt sRNAs were shown to be selectively processed from a similar number of MITE-derived miRNA precursor transcripts in rice, with the observed MITE-derived miRNA size heterogeneity proposed to reflect the wide spectrum of evolutionary intermediates as individual MITEs transition from TE loci to *MIR* genes [100]. In a subsequent study [168], where the authors relaxed their initial stringent selection parameters to remove any putative miRNAs originating from TEs or from other classes of DNA repeats to allow for the inclusion of such TE-derived miRNAs, 53 of the 80 (~66%) TE-derived miRNAs identified were determined to have originated from MITEs. Via the analysis of the accumulation of MITE-derived miRNAs in the rice mutant lines *dcl1*, *dcl3a,* and *rdr2*, all 21- and 24-nt MITE-derived miRNAs identified in this study were subsequently shown to require *Osa*-DCL1 and *Osa*-DCL3A for their production, respectively [168]. Further evidence that all identified MITE-derived miRNAs were indeed ‘real’ miRNAs regardless of their length, and not incorrectly annotated rasiRNAs, was provided via the demonstration that the abundance of each MITE-derived miRNA remained at its wild-type equivalent level in the rice *rdr2* mutant and that all 21-nt MITE-derived miRNAs were loaded by *Osa*-AGO1 and not by *Osa*-AGO4: the primary AGO effector protein of the rasiRNA-directed RdDM pathway [168].

Alignment of the precursor transcripts of the rice miRNAs, *Osa*-miR806a, *Osa*-miR812a, *Osa*-miR812b, *Osa*-miR812f, *Osa*-miR812h, *Osa*-miR812i, *Osa*-miR812j, *Osa*-miR814a, *Osa*-miR814b, *Osa*-miR818e, *Osa*-miR1850, *Osa*-miR1862d, *Osa*-miR1862e, *Osa*-miR1867, *Osa*-miR1868, *Osa*-miR1884a, and *Osa*-miR1884b, has repeatedly shown significant sequence similarities to various MITE species [99,100,146,160,169,170]. Additional sequence analysis further revealed that for some of these MITE-derived miRNAs, including *Osa*-miR1850, *Osa*-miR1867, *Osa*-1884a, and *Osa*-1884b, a 21-nt variant offset by several nt at the 5′ end of the annotated and predominant 24-nt species of each miRNA is also processed from the same precursor transcript [169]. Interestingly, the offset processing of 21-nt MITE-derived miRNA variants was shown to result in each variant containing a 5′ terminal uracil residue, to generate miRNAs of the preferred size, and with the 5′ terminal nt composition preference, to direct their selective loading by *Osa*-AGO1, as elegantly demonstrated by Ou-Yang and colleagues (2013) via immunoprecipitation of *Osa*-AGO1 and profiling of its miRNA payload [169]. Finally, and as part of their extensive bioinformatic analyses, the authors additionally showed that the MITEs from which MITE-derived miRNAs have originated, have preferentially transposed into the 5′ and 3′ UTRs of numerous protein-coding genes [169]. Therefore, when considered together with their profiling of the miRNA payload of immunoprecipitated *Osa*-AGO1, the presented findings infer that members of this MITE-derived miRNA cohort may potentially regulate the expression of each MITE-containing protein-coding gene via directing an AGO1-catalyzed target transcript cleavage mode of gene expression regulation, as has been demonstrated for conserved miRNAs in rice [171].

The miRBase database lists 22 members for the rice *MIR812* gene family, including *Osa*-*MIR812a* through to *Osa-MIR812v*, with a *Stowaway*-like MITE identified to have played a central role in the considerable expansion of this *MIR* gene family in rice [146,160]. A more recent study [170] identified *Osa*-miR812w, and via precursor transcript sequence analyses, it was shown that this additional *Osa-MIR812* family member also resulted from the transposition of a *Stowaway*-like MITE. Northern blot hybridization analysis was used to demonstrate *Osa*-miR812w accumulation in all 11 of the species of cultivated rice assessed; however, of the seven wild rice species also analyzed via northern blotting, the *Osa*-miR812w sRNA was shown to accumulate to detectable levels in the five wild rice species with an AA genome and not in species with either a CC (*O. latifolia*; broadleaf rice) or CCDD (*O. officinalis*; South East Asian rice) genome [170]. Identification of the *Osa-MIR812w* locus in the wild rice ancestors of modern cultivated *Oryza* species suggests that this *MIR* gene was conserved during the domestication of rice, including African rice (*O. glaberrima*) and Asian rice (*O. sativa*). Like many other rice MITE-derived miRNAs, the *Osa*-miR812w sRNA is 24-nt in length, with the assessment of *Osa*-miR812w accumulation in the rice *dcl1*, *dcl3a*, *dcl4,* and *rdr2* mutant backgrounds revealing that only *Osa*-DCL3A is required for *Osa*-miR812w production [170]. Immunoprecipitation and profiling of the sRNA species loaded by *Osa*-AGO1a, *Osa*-AGO1b, *Osa*-AGO4a, and *Osa*-AGO4b have previously shown that the 24-nt *Osa*-miR812w sRNA is preferentially loaded by *Osa*-AGO4b [171,172]. Computational analyses have further revealed that 36 of the 39 (~92%) protein-coding genes that harbor copies of the same *Stowaway*-like MITE from which *Osa*-miR812w is derived are inserted in either the 5′ (*n =* 6) or 3′ (*n =* 30) UTR of these protein-coding genes [170]. Campo et al. (2021) went on to demonstrate that *Osa*-miR812w regulates the expression of three genes that house *Stowaway*-MITE insertions in their UTRs, including the *1-AMINOCYCLOPROPANE-1-CARBOXYLATE OXIDASE3* (*ACO3*), *CALCINEURIN B LIKE* (*CBL*)-*INTERACTING SERINE-THREONINE PROTEIN KINASE10* (*CIPK10*), and *LEUCINE RICH REPEAT-CONTAINING PROTEIN* (LRR) loci, via exclusively directing highly localized (within 80-nt of the *Osa*-miR812w target site) DNA methylation [170]. This finding elegantly demonstrates the uniqueness of the miRNA expression modules of rice MITE-derived miRNAs, specifically: MITE-derived miRNA precursor transcripts are processed by *Osa*-DCL3a to produce 24-nt miRNAs, which are subsequently loaded by *Osa*-AGO4b to direct *Osa*-AGO4b-catalyzed target gene expression regulation via highly localized DNA methylation.

## 8. Concluding Remarks

For several decades following their initial discovery, TEs, like other classes of DNA repeats, were largely viewed as a form of ‘junk,’ ‘selfish,’ or even ‘parasitic’ DNA. However, this view was initially challenged due to the vast volume of total nuclear genome space occupied by TEs in many angiosperm species: a considerable genome volume that strongly hinted at an unknown, yet highly important, genetic function. More recently, and especially over the last decade, profiling of the expansive populations of small and long ncRNA species derived from TE-rich regions of angiosperm nuclear genomes, has provided extensive evidence of the significant contribution that TEs have made to the overall regulation of global gene expression as part of the continuing evolution of angiosperm nuclear genomes. Moreover, TE activation, proliferation, and movement can have a profound effect on gene expression and/or function, with TE transposition into the (1) regulatory region of a protein-coding gene potentially altering the level, temporality, stage of development, and/or spatial pattern of gene expression, or (2) coding region of a protein-coding gene potentially resulting in the synthesis of either a dysfunctional or completely non-functional protein product. Therefore, to combat the potentially harmful consequences of TE activation, proliferation, and movement, angiosperms have developed an elegant molecular strategy to keep the majority of the TE species contained in their nuclear genomes in an inactive state. Namely, the rasiRNA-directed RdDM pathway, a mechanism of RNA silencing specific to plants, forms a highly effective molecular mechanism employed by the angiosperms to repress TE activation, proliferation, and/or movement.

Due to their preference to transpose within gene-rich regions of angiosperm nuclear genomes, together with their short length, and the presence of TIR and TSD sequences at their 5′ and 3′ termini, the MITE species of Type II TEs have formed a rich source of new and novel *MIR* genes in many angiosperm species. More specifically, MITE transposition into the body of a protein-coding gene, or immediately adjacent to a protein-coding gene, provides this species of nonautonomous TE with the transcriptional activity required for its expression. Post-transcription, the TIR at each end of the resulting ncRNA affords the transcript the ability to fold and form a dsRNA structure that is recognized by DCL enzymes, the core pieces of protein machinery in RNA silencing pathways. DCL processing of MITE-derived dsRNAs generates 21- or 24-nt miRNAs, which, after their maturation, are used by AGO proteins as sequence specificity guides to direct gene expression changes at either the transcriptional (i.e., AGO4-loaded 24-nt MITE-derived miRNAs direct RdDM of target gene sequences) or posttranscriptional level (i.e., AGO1-loaded 21-nt MITE-derived miRNAs direct mRNA cleavage or translational repression of target gene transcripts). Therefore, although the angiosperms have developed elegant molecular strategies to control TE activity, MITEs have evolved their own genetic properties to avoid the RNA silencing strategies of a plant, namely, the adoption of dsRNA folding structures that mimic those of the precursor transcripts of conserved angiosperm miRNAs, which has enabled the MITE species of Type II TEs to make a considerable contribution to the total *MIR* gene population of numerous angiosperm species. Armed with our present knowledge of the major species of MITEs that have contributed most heavily to *MIR* gene evolution in the angiosperms, it may well be time to revisit the numerous genomic and transcriptomic datasets generated as part of the early molecular characterization of the angiosperms. Interrogation of these early datasets would lead to the identification of additional MITE-derived miRNAs, miRNAs that would have been excluded from the original bioinformatic analyses due to their TE origins. Such an endeavor would provide a more accurate determination of the true contribution the MITE class of Type II TE has made to the global miRNA landscape of the angiosperms.

## Figures and Tables

**Figure 1 plants-12-01101-f001:**
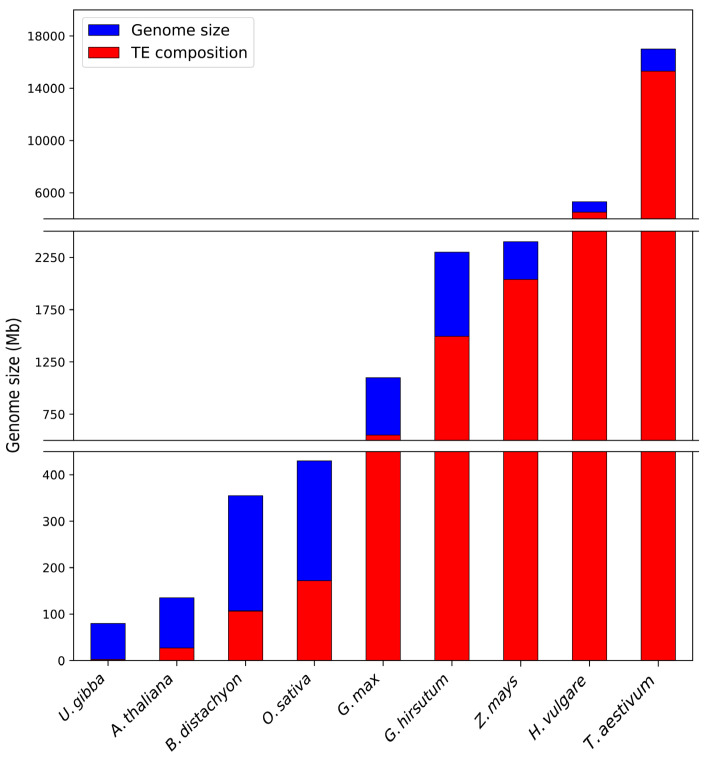
Comparison of total genome size with TE load across representative angiosperms. Comparison of the total size (megabases; Mb) of the nuclear genomes of nine angiosperm species with the total TE load (presented as a Mb value converted from a % value of total nuclear genome size) of each assessed species revealed a strong positive correlation between increased genome size and increased TE load. Data on nuclear genome size and TE load of the nine angiosperm species were obtained from [6,7,8,9,10,11,12,13].

**Figure 2 plants-12-01101-f002:**
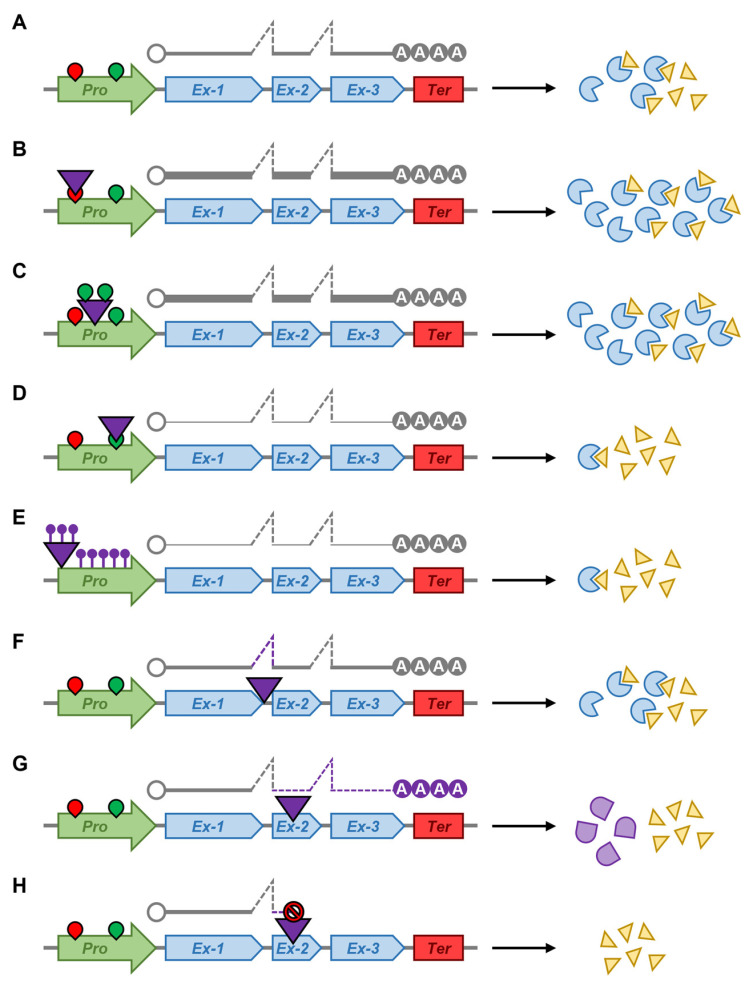
Consequences of TE insertion into the regulatory region or coding sequence of an angiosperm gene. (**A**) Schematic representation of the general DNA-based features of an angiosperm protein-coding gene, including a promoter region (*Pro*; green-shaded arrow), three exons (*Ex-1*, *Ex-2*, *Ex-3*; blue-shaded pentagons), terminator (*Ter*; red-shaded rectangle), the transcribed mRNA (solid (exons) and dashed (introns) grey line), the encoded protein product (an enzyme) of the gene (blue-shaded partial circles), and its specific substrate (yellow-shaded triangles). (**B**) TE insertion (purple-shaded triangle) into an inhibitory *cis*-regulatory element (red-shaded teardrop) in the promoter region of a protein-coding gene can enhance the expression level of the gene. (**C**) A TE can house its own *cis*-regulatory elements (enhancers; green-shaded tear drops), and upon promoter region insertion, protein-coding gene expression can be altered (elevated gene expression example provided). (**D**) TE insertion into an enhancing *cis*-regulatory element (green-shaded teardrop) can reduce protein-coding gene expression. (**E**) Insertion of an epigenetically marked TE (purple-shaded paddles) into the promoter region (or coding sequences) of a protein-coding gene can mark the gene as a new target sequence for epigenetic modification (i.e., DNA methylation) to repress gene expression. (**F**) TE insertion into an intron may have no influence on the overall level of expression of the gene. (**G**) TE insertion into the coding region of a protein-coding gene can result in a frameshift mutation (dashed purple line), which will in turn result in the production of an incorrectly folded and therefore dysfunctional protein product (purple-shaded ‘chord’ symbol). (**H**) Similarly, TE insertion into the coding region of a protein-coding gene can introduce a premature stop codon (red-shaded ‘not allowed’ symbol) to the reading frame of the gene, which would result in no functional protein being produced.

**Figure 3 plants-12-01101-f003:**
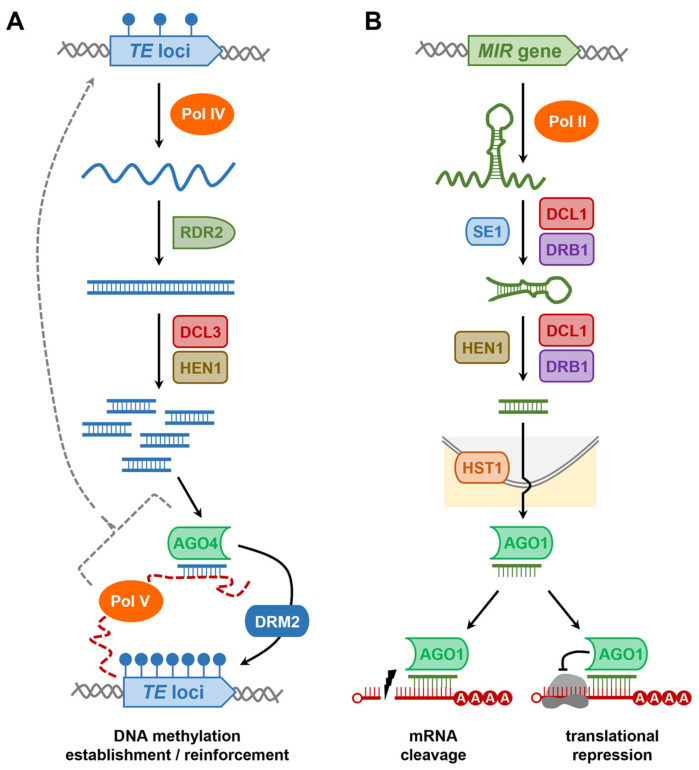
The core components and stages of the canonical RNA-directed DNA methylation and microRNA pathways of angiosperms. (**A**) RdDM pathway: methylated *TE* loci (blue-shaded pentagon with solid blue lollipops) are recognized by Pol IV as a template for ncRNA (wavy blue line) transcription. RDR2 converts the TE-derived ncRNA into a perfect dsRNA molecule (blue long horizontal ladder), which is subsequently processed into 24-nt rasiRNA/rasiRNA* duplexes (blue short horizontal ladders) by DCL3, with the 3′ end of each duplex strand methylated by HEN1. Following duplex strand separation, rasiRNA guide strand (blue inverted comb) loaded AGO4 forms the catalytic core of the RITS complex, and via interaction with Pol V-generated ncRNAs (red colored dashed line) and a suite of DNA methyltransferases (including DRM2) and other histone modification enzymes, directs the establishment of DNA methylation (black solid line) or the reinforcement of DNA methylation (grey dashed line) of complementary TE loci to keep this class of DNA repeat in an inactive state. (**B**) miRNA pathway: a long ncRNA is transcribed from a *MIR* gene (green-shaded pentagon) by Pol II, which folds to form a pri-miRNA (dark green wavy and looped line). SE1 binds the pri-miRNA and delivers it to DCL1/DRB1 for processing to produce the pre-miRNA (dark green stem-loop). DCL1/DRB1 further processes the pre-miRNA to produce the miRNA/miRNA* duplex (dark green short horizontal ladder), with HEN1 methylating the 3′ end of each duplex strand. Following duplex strand separation, the miRNA guide strand (dark green inverted comb) is exported to the cytoplasm via either a HST1-dependent or -independent mechanism, where it is loaded by AGO1 and used as a sequence specificity guide to repress the expression of highly complementary target gene transcripts. In the angiosperms, miRNA target gene expression is regulated by either mRNA cleavage or translational repression mode of RNA silencing.

**Figure 4 plants-12-01101-f004:**
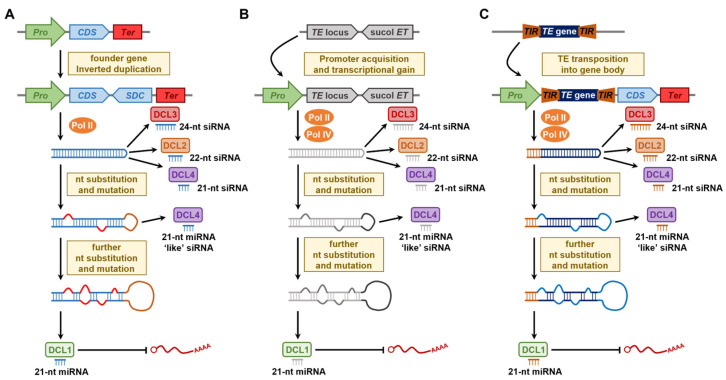
Angiosperm *MICRORNA* gene evolution Models 1, 2, and 3. (**A**) Model 1—Target Gene Duplication: the newly duplicated copy of the founder gene inserts into the existing gene body of the founder gene in an inverted orientation. Post-transcription by Pol II, the new transcript folds to form a perfect dsRNA molecule, which is processed into 21-, 22-, and 24-nt siRNAs by DCL4, DCL2, and DCL3, respectively. Additional nt substitutions and mutations modify the dsRNA folding structure to transition it to a structure similar to that of a newly-evolved miRNA precursor, which now forms a DCL4 processing substrate for 21-nt miRNA-like siRNA production. Additional nt alterations would further transform the structure of the dsRNA to match that of a precursor of a conserved miRNA, thereby solely forming a DCL1 processing substrate for 21-nt miRNA production. (**B**) Model 2—Random Hairpin RNA Forming Structures: any genomic sequence with the ability to form a hpRNA post-transcription (TE inverted repeat example provided) acquires transcriptional activity via the obtainment of a protein-coding gene promoter or the transcription regulating elements of an autonomous TE, which would enable its use as a template for transcription by either Pol II or Pol IV. (**C**) Model 3—Transposed TE-derived Model: After the gain of transcriptional activity (transposition into a protein-coding gene body example provided), the TE forms a template for transcription by either Pol II or Pol IV. (**B**,**C**) As outlined in Model 1, *MIR* gene evolution Models 2 and 3 also propose that the initially folded dsRNA will form a processing substrate for DCL2, DCL3, and DCL4 to produce 22-, 24-, and 21-nt siRNAs, respectively. Subsequent nt substitutions, mutations, and/or other sequence alterations will transition the dsRNA to a processing substrate for DCL4-directed production of 21-nt miRNA-like siRNAs and then to a stem-loop structured dsRNA, which is solely recognized for processing by DCL1 to produce a 21-nt TE-derived miRNA.

**Figure 5 plants-12-01101-f005:**
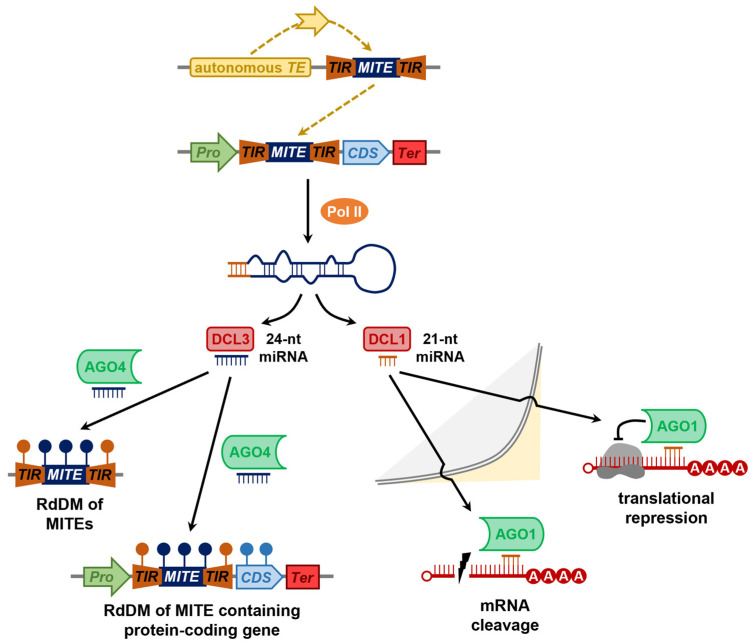
The MITE-derived miRNA pathway in angiosperms. MITEs preferentially transpose into gene-rich regions of angiosperm nuclear genomes following their initial excitation by a transposase enzyme encoded by an autonomous Type II TE. MITE transposition into the body of a protein-coding gene results in the MITE being included for use as a template for transcription by Pol II. The MITE-derived ncRNA folds to form a dsRNA stem-loop structure that very closely mimics the dsRNA folding structures adopted by the pri-miRNA and pre-miRNA precursors of conserved angiosperm miRNAs. Depending on the angiosperm species under investigation, MITE-derived dsRNAs have been shown to be predominantly processed by DCL3 to produce a 24-nt miRNA, which, post-maturation, is loaded by AGO4. AGO4 uses the loaded MITE-derived miRNA as a sequence specificity determinant to guide highly localized RdDM of other MITEs of similar sequence composition or of protein-coding genes that harbor a complementary MITE insertion. Alternatively, it has been shown in other angiosperm species that MITE-derived dsRNAs are processed by DCL1 to produce 21-nt MITE-derived miRNAs. Post its maturation in the nucleus and its subsequent exportation to the cytoplasm, AGO1 uses the MITE-derived miRNA as a sequence specificity determinant to control the expression of target genes at the posttranscriptional level via either directing a mRNA cleavage or translational repression mode of RNA silencing.

## Data Availability

Not applicable.

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
