# Peer review of "Miniature Inverted-Repeat Transposable Elements: Small DNA Transposons That Have Contributed to Plant MICRORNA Gene Evolution"

_plants, 2023, doi:10.3390/plants12051101_

Round 1

Reviewer 1 Report

The presented review is devoted to the transposable elements and their relationship with miRNAs. The manuscript contains fairly extensive and interesting information. However, it seems that such extensive material is not sufficiently illustrated.  Also many sentences are overloaded which will make it difficult to understand for many.

Line 117: “elegant molecular mechanisms to 117 mitigate the negative consequences of TE activity” Please list these mechanisms.

Line 204-208: Please provide a table with the main MATE superfamilies and their characteristics.

Author Response

Dear Editor,

On behalf of the authorship team, I would like to thank the three Reviewers for their highly positive appraisal of our review article, and we (the authors) are very appreciative of the helpful and constructive comments made by Reviewers 1, 2 and 3. Please see below our responses to the constructive comments made by the three Reviewers.

We have uploaded a revised manuscript version to the system as requested. Below, and for your and the reviewers’ ease of reference, we have stated the new page / line numbers of the revised manuscript where we have attempted to address each comment of each reviewer. Page / line number changes are the result of uploading a ‘tracked changes’ file format of our revised review article as requested.

If any additional clarification is required regarding the changes made as part of revising the review article, then please do not hesitate in contacting either myself, or co-correspondence author, Dr. Joseph Pegler.

Kind regards,

Andrew Eamens (on behalf of the authorship team)  

Reviewer #1

Comments and Suggestions for Authors

The presented review is devoted to the transposable elements and their relationship with miRNAs. The manuscript contains fairly extensive and interesting information. However, it seems that such extensive material is not sufficiently illustrated.  Also many sentences are overloaded which will make it difficult to understand for many.

  • We thank Reviewer 1 for their suggestion that we have not sufficiently illustrated much of the extensive and interesting information covered in our review article. However, we have included 5 Figures in our review article which very clearly illustrate the central themes covered within the review article. The authors are therefore of the opinion that we have provided adequate illustration of the central themes of the review, and that the inclusion of additional Figures would not provide any additional impact or benefit to the reader.
  • Further, Reviewers 2 and 3 have commended the authors on their writing style. Therefore, based on the highly positive comments of Reviewers 2 and 3, restructuring of the writing style of the sentences contained within the text of this review article does not appear to be warranted.

Line 117: “elegant molecular mechanisms to 117 mitigate the negative consequences of TE activity” Please list these mechanisms.

  • We thank Reviewer 1 for raising this comment. However, at line numbers 141-144 (we have completely rewritten the Abstract which changes the line numbers in the ‘track changes’ version of our revised review article) we are only making mention of the elegant molecular mechanisms used by angiosperms to repress TE activity at this early stage of the review. Extensive additional information is first required to completely describe TEs before subsequently addressing how angiosperms keep TEs in a repressed state. If we were to describe the RdDM mechanism of TE activity repression earlier in the review article, then the entire structure of the review article would no longer be appropriate. The authors are therefore of the opinion that the present structure and order of the various sections of the review article is appropriate as it forms an ordered structure which will be most readily digested by the readers of the article.

Line 204-208: Please provide a table with the main MATE superfamilies and their characteristics.

  • We thank Reviewer 1 for this suggestion. However, the type of information which would be included in the requested Table listing MITE characteristics (e.g., their size and sequence composition) has been covered elsewhere previously by others. Further, this degree of requested information is outside of the scope of this review article as it would not provide any additional insight/information to the review considering the other major foci of the review article. The authors are therefore of the opinion that the inclusion of such a Table would not add any additional impact / insight to the review.

In addition, the authors would like to take this opportunity to thank Reviewer 1 for their positive review of our submitted review article. We have made a number of changes to our original submission for its improvement based of the comments of all three reviewers, with the list below identifying the specific changes made in the revised version of our manuscript for convenience of reference for Reviewer 1.

Revisions include:

  • Page 1, lines 15-35: A new Abstract has been written to attempt to more accurately describe the ‘core motifs’ of the review article (as suggested by Reviewer 3).
  • Page 2, lines 55-58: We have reduced the number of Keywords from 9 to 7 (as suggested by Reviewer 3).
  • Page 4, lines 109, 114, 116-118, 120-122: multiple text changes have been made to the Figure 2 legend for consistency throughout the text (primarily ‘gene’ has been changed to ‘protein-coding gene’ to align the wording of the legend to that of the section text).
  • Page 6, line 213: the word ‘internal’ has been added to give ‘internal nucleotide composition’ to provide more specific information.
  • Page 7, line 240: the word ‘total’ has been added to give ‘total MITE copy number’ to be more specific.
  • Page 7, line 267: the word ‘surrounding’ was replaced with ‘adjacent to’ to improve sentence wording.
  • Page 7, lines 280-281: the word ‘early’ has been changed to ‘initial’ to improve sentence wording.
  • Page 9, lines 351-352: the words ‘the source for the production of the’ have been replaced with ‘processed to produce a novel member of the’ to improve the overall wording of the associated sentence.
  • Page 12, line 481: wording changed from ‘ a MITE-derived ncRNA transcript’ to ‘the MITE-derived ncRNA’ for consistency throughout the text.
  • Page 12, line 487: the word ‘stressors’ has been changed to ‘stresses’ as it is more appropriate to the field.
  • Page 13, line 549: the statement ‘: referred to as newly-evolved miRNAs from herein’ has been added to ensure terminology consistency.
  • Page 14, line 595: ‘closely-species’ has been corrected to ‘closely-related’ for text improvement.
  • Page 14, line 605: the text ‘also appears’ has been replaced with ‘has also been shown’ for wording improvement.
  • Page 14, line 638: ‘RNA’ has been corrected to ‘ncRNA’ for text improvement.
  • Page 15, lines 663-665: the text ‘miRNA-like siRNA production by DCL4 (such as the DCL4 processing requirements of’ has been changed to ‘the production of a reduced number of miRNA-like siRNAs by DCL4 (such as the DCL4 processing patterns documented for’ for sentence wording improvement.
  • Page 16, line 718: the word ‘hairpin’ has been corrected to ‘hpRNA’.
  • Page 17, line 765: ‘dicotyledonous’ was changed to ‘dicot’ as the abbreviation has already been stated.
  • Page 18, line 818: the text ‘were demonstrated to’ has been deleted for improved sentence wording / structure.
  • Page 19, line 822: the words ‘Type II’ has been added as a correction.
  • Page 21, line 940: wording ‘putative miRNAs identified to a MITE’ has been changed to ‘putative miRNAs identified in this species to a MITE origin’ to correct the wording of this sentence section.
  • Page 22, line 980: ‘proteins’ has been corrected to ‘protein’.

Page 24, lines 1077-1085: Additional sentences have been added to the final paragraph of the Concluding remarks section of the revised review article to attempt to summarise the importance of our findings to research (as suggested by Reviewer 3).

Reviewer 2 Report

The manuscript by Pegler et al. discusses the role of MITEs in plant genome evolution. Based on recent literature, the authors provide information about transposable elements in various plants and especially for MITEs. Further, they describe the mechanism used by plants to repress TEs and how MITEs avoid the RdDM repression. Finally, they correlate this mechanism with the MIR gene evolution. The manuscript is well written and comprehensive, enriched with descriptive figures of the models discussed in the text. Therefore, I believe that the paper should be accepted for publication. 

Author Response

Dear Editor,

On behalf of the authorship team, I would like to thank the three Reviewers for their highly positive appraisal of our review article, and we (the authors) are very appreciative of the helpful and constructive comments made by Reviewers 1, 2 and 3. Please see below our responses to the constructive comments made by the three Reviewers.

We have uploaded a revised manuscript version to the system as requested. Below, and for your and the reviewers’ ease of reference, we have stated the new page / line numbers of the revised manuscript where we have attempted to address each comment of each reviewer. Page / line number changes are the result of uploading a ‘tracked changes’ file format of our revised review article as requested.

If any additional clarification is required regarding the changes made as part of revising the review article, then please do not hesitate in contacting either myself, or co-correspondence author, Dr. Joseph Pegler.

Kind regards,

Andrew Eamens (on behalf of the authorship team)  

The authors would like to take this opportunity to thank Reviewer 2 for the highly positive review of our submitted review article supporting its acceptance for publication. Although Reviewer 2 has not requested any modification to our original submission, we list below the specific changes made in the revised version of our manuscript for convenience of reference for Reviewer 2.

Revisions include:

  • Page 1, lines 15-35: A new Abstract has been written to attempt to more accurately describe the ‘core motifs’ of the review article (as suggested by Reviewer 3).
  • Page 2, lines 55-58: We have reduced the number of Keywords from 9 to 7 (as suggested by Reviewer 3).
  • Page 4, lines 109, 114, 116-118, 120-122: multiple text changes have been made to the Figure 2 legend for consistency throughout the text (primarily ‘gene’ has been changed to ‘protein-coding gene’ to align the wording of the legend to that of the section text).
  • Page 6, line 213: the word ‘internal’ has been added to give ‘internal nucleotide composition’ to provide more specific information.
  • Page 7, line 240: the word ‘total’ has been added to give ‘total MITE copy number’ to be more specific.
  • Page 7, line 267: the word ‘surrounding’ was replaced with ‘adjacent to’ to improve sentence wording.
  • Page 7, lines 280-281: the word ‘early’ has been changed to ‘initial’ to improve sentence wording.
  • Page 9, lines 351-352: the words ‘the source for the production of the’ have been replaced with ‘processed to produce a novel member of the’ to improve the overall wording of the associated sentence.
  • Page 12, line 481: wording changed from ‘ a MITE-derived ncRNA transcript’ to ‘the MITE-derived ncRNA’ for consistency throughout the text.
  • Page 12, line 487: the word ‘stressors’ has been changed to ‘stresses’ as it is more appropriate to the field.
  • Page 13, line 549: the statement ‘: referred to as newly-evolved miRNAs from herein’ has been added to ensure terminology consistency.
  • Page 14, line 595: ‘closely-species’ has been corrected to ‘closely-related’ for text improvement.
  • Page 14, line 605: the text ‘also appears’ has been replaced with ‘has also been shown’ for wording improvement.
  • Page 14, line 638: ‘RNA’ has been corrected to ‘ncRNA’ for text improvement.
  • Page 15, lines 663-665: the text ‘miRNA-like siRNA production by DCL4 (such as the DCL4 processing requirements of’ has been changed to ‘the production of a reduced number of miRNA-like siRNAs by DCL4 (such as the DCL4 processing patterns documented for’ for sentence wording improvement.
  • Page 16, line 718: the word ‘hairpin’ has been corrected to ‘hpRNA’.
  • Page 17, line 765: ‘dicotyledonous’ was changed to ‘dicot’ as the abbreviation has already been stated.
  • Page 18, line 818: the text ‘were demonstrated to’ has been deleted for improved sentence wording / structure.
  • Page 19, line 822: the words ‘Type II’ has been added as a correction.
  • Page 21, line 940: wording ‘putative miRNAs identified to a MITE’ has been changed to ‘putative miRNAs identified in this species to a MITE origin’ to correct the wording of this sentence section.
  • Page 22, line 980: ‘proteins’ has been corrected to ‘protein’.
  • Page 24, lines 1077-1085: Additional sentences have been added to the final paragraph of the Concluding remarks section of the revised review article to attempt to summarise the importance of our findings to research (as suggested by Reviewer 3).

Reviewer 3 Report

Title: Miniature inverted repeat transposable elements: small DNA transposons that have contributed to plant MicroRNA gene evolution

The review is logically organized and reports the actual knowledge for the selected field of interest. 

Please, consider some small notes to improve the manuscript.

In abstract, be more specific for the core motif of the review.

Keywords - too many are used.

Conclusion - summarize the importance of your findings for research.

Author Response

Dear Editor,

On behalf of the authorship team, I would like to thank the three Reviewers for their highly positive appraisal of our review article, and we (the authors) are very appreciative of the helpful and constructive comments made by Reviewers 1, 2 and 3. Please see below our responses to the constructive comments made by the three Reviewers.

We have uploaded a revised manuscript version to the system as requested. Below, and for your and the reviewers’ ease of reference, we have stated the new page / line numbers of the revised manuscript where we have attempted to address each comment of each reviewer. Page / line number changes are the result of uploading a ‘tracked changes’ file format of our revised review article as requested.

If any additional clarification is required regarding the changes made as part of revising the review article, then please do not hesitate in contacting either myself, or co-correspondence author, Dr. Joseph Pegler.

Kind regards,

Andrew Eamens (on behalf of the authorship team) 

Reviewer #3

Comments and Suggestions for Authors

Title: Miniature inverted repeat transposable elements: small DNA transposons that have contributed to plant MicroRNA gene evolution

The review is logically organized and reports the actual knowledge for the selected field of interest. 

  • The authorship team wish to thank Reviewer 3 for their positive review of our article. The authors also wish to thank Reviewer 3 for identifying the three (3) areas which could be improved on in a revised version of our originally submitted review article. Please see our responses to the 3 points raised by Reviewer 3 below: the page and line numbers for the requested changes are provide for the convenience of the reviewer.

Please, consider some small notes to improve the manuscript.

In abstract, be more specific for the core motif of the review.

  • The authors thank Reviewer 3 for identifying the lack of focus in the Abstract of our original submission. We have completely modified the Abstract in the revised version of our manuscript to address this reviewer concern. Please see page 1, lines 15 – 35, of the revised manuscript version (tracked changes file format submitted as requested by the journal).

Keywords - too many are used.

  • The Microsoft Word template provided by Plants states ‘List three to ten pertinent keywords specific to the article yet reasonably common within the subject discipline’. Originally, we had used 9 Keywords, however, to address this reviewer concern, the Keywords ‘nuclear genome’ and ‘small-interfering RNA (siRNA)’ have been removed to bring the total number of Keywords used in the revised version of our manuscript to 7. In addition, Keyword ‘small RNA’ has been changed to ‘repeat-associated small-interfering RNA (rasiRNA) to more accurately describe one of the two major small RNA species discussed in detail in the review article. Please see page 2, lines 55 – 58, of the revised manuscript version (tracked changes file format submitted as requested by the journal).

Conclusion - summarize the importance of your findings for research.

  • The authors thank Reviewer #3 for identifying that we had not adequately stated the importance to research the major foci of our review article. At the end of the Concluding remarks section of our review article we have added several new sentences to attempt to more readily communicate to the reader the importance of the major findings of our review with respect to future research undertakings in this area. Please see page 24, lines 1074 – 1082, of the revised manuscript version (tracked changes file format submitted as requested by the journal).

For the convenience of Reviewer 3, we provide a list below of the other changes we have additionally made to our originally submitted manuscript as part of the revision process. These include:

Revisions include:

  • Page 1, lines 15-35: A new Abstract has been written to attempt to more accurately describe the ‘core motifs’ of the review article (as suggested by Reviewer 3).
  • Page 2, lines 55-58: We have reduced the number of Keywords from 9 to 7 (as suggested by Reviewer 3).
  • Page 4, lines 109, 114, 116-118, 120-122: multiple text changes have been made to the Figure 2 legend for consistency throughout the text (primarily ‘gene’ has been changed to ‘protein-coding gene’ to align the wording of the legend to that of the section text).
  • Page 6, line 213: the word ‘internal’ has been added to give ‘internal nucleotide composition’ to provide more specific information.
  • Page 7, line 240: the word ‘total’ has been added to give ‘total MITE copy number’ to be more specific.
  • Page 7, line 267: the word ‘surrounding’ was replaced with ‘adjacent to’ to improve sentence wording.
  • Page 7, lines 280-281: the word ‘early’ has been changed to ‘initial’ to improve sentence wording.
  • Page 9, lines 351-352: the words ‘the source for the production of the’ have been replaced with ‘processed to produce a novel member of the’ to improve the overall wording of the associated sentence.
  • Page 12, line 481: wording changed from ‘ a MITE-derived ncRNA transcript’ to ‘the MITE-derived ncRNA’ for consistency throughout the text.
  • Page 12, line 487: the word ‘stressors’ has been changed to ‘stresses’ as it is more appropriate to the field.
  • Page 13, line 549: the statement ‘: referred to as newly-evolved miRNAs from herein’ has been added to ensure terminology consistency.
  • Page 14, line 595: ‘closely-species’ has been corrected to ‘closely-related’ for text improvement.
  • Page 14, line 605: the text ‘also appears’ has been replaced with ‘has also been shown’ for wording improvement.
  • Page 14, line 638: ‘RNA’ has been corrected to ‘ncRNA’ for text improvement.
  • Page 15, lines 663-665: the text ‘miRNA-like siRNA production by DCL4 (such as the DCL4 processing requirements of’ has been changed to ‘the production of a reduced number of miRNA-like siRNAs by DCL4 (such as the DCL4 processing patterns documented for’ for sentence wording improvement.
  • Page 16, line 718: the word ‘hairpin’ has been corrected to ‘hpRNA’.
  • Page 17, line 765: ‘dicotyledonous’ was changed to ‘dicot’ as the abbreviation has already been stated.
  • Page 18, line 818: the text ‘were demonstrated to’ has been deleted for improved sentence wording / structure.
  • Page 19, line 822: the words ‘Type II’ has been added as a correction.
  • Page 21, line 940: wording ‘putative miRNAs identified to a MITE’ has been changed to ‘putative miRNAs identified in this species to a MITE origin’ to correct the wording of this sentence section.
  • Page 22, line 980: ‘proteins’ has been corrected to ‘protein’.
  • Page 24, lines 1077-1085: Additional sentences have been added to the final paragraph of the Concluding remarks section of the revised review article to attempt to summarise the importance of our findings to research (as suggested by Reviewer 3).